# Robust switches in thalamic network activity require a timescale separation between sodium and T-type calcium channel activations

Kathleen Jacquerie[ID]*, Guillaume Drion[ID]

Department of Electrical Engineering and Computer Science, University of Liege, Liege, Belgium

* kathleen.jacquerie@uliege.be

## Abstract

Switches in brain states, synaptic plasticity and neuromodulation are fundamental processes in our brain that take place concomitantly across several spatial and timescales. All these processes target neuron intrinsic properties and connectivity to achieve specific physiological goals, raising the question of how they can operate without interfering with each other. Here, we highlight the central importance of a timescale separation in the activation of sodium and T-type calcium channels to sustain robust switches in brain states in thalamic neurons that are compatible with synaptic plasticity and neuromodulation. We quantify the role of this timescale separation by comparing the robustness of rhythms of six published conductance-based models at the cellular, circuit and network levels. We show that robust rhythm generation requires a T-type calcium channel activation whose kinetics are situated between sodium channel activation and T-type calcium channel inactivation in all models despite their quantitative differences.

## Author summary

Our brain is constantly processing information either from the environment to quickly react to incoming events or learning from experience to shape our memory. These brain states translate a collective activity of neurons interconnected via synaptic connections. Here, we focus on the thalamic network showing a transition from an active to an oscillatory mode at the population level, reverberating a switch from tonic to bursting mode at the cellular level. We are questioning how these activity fluctuations can be robustly modeled despite synaptic plasticity affecting the network configuration and the presence of neuromodulators affecting neuron intrinsic properties. To do so, we investigate six conductance-based models and their ability to reproduce activity switches at the cellular, circuit and population levels. We highlight that the robustness requires the timescale separation between the fast activation of sodium channels compared to the slow activation of T-type calcium channels. Our results show that this kinetics difference is not a

**Data Availability Statement:** Julia and matlab code files are freely available at http://www.montefiore.ulg.ac.be/~guilldrion/Files/Jacquerie2021_codes.zip and https://osf.io/sth4d/.

Models are described in the Supporting information Files.

**Funding:** K.J has received funding from the Belgian National Fund for Scientific Research - FNRS (ID application: 34959817, https://www.frs-fnrs.be/en/). The funders had no role in study design, data collection and analysis, decision to publish, or preparation of the manuscript.

**Competing interests:** The authors have declared that no competing interests exist.

computational detail but rather makes a model suitable and robust to study the interaction between switches in brain states, synaptic plasticity and neuromodulation.

## Introduction

Animal performance relies on their ability to quickly process, analyze and react to incoming events, as well as to learn from experience to constantly increase their knowledge about the environment. This information processing is shaped by fluctuations in rhythmic neuronal activities at the cellular and population levels, each defining *brain states* [1, 2]. These activities are recognizable by spatiotemporal signatures of the mean-field activity of large neuronal populations. Switches between these brain states can be fast and localized, such as for example those observed in different brain areas prior to movement initiation [3], or global and long-lasting, such as those observed during the wake-sleep transition [4, 5].

In the thalamus, cellular recordings reveal a firing pattern transition during wake-sleep cycles. The thalamic neurons switch from a regular spiking mode to a bursting mode [6, 7]. This feature is not strictly restricted to sleep but can also appear during an awake and vigilant behavior [8–12]. At the population level, the change in cadence is remarkable; the mean-field activity rapidly switches from an active state to an oscillatory state [13, 14].

The mechanisms governing rhythm fluctuations are poorly understood. This research problem is difficult to solve since it involves phenomena occurring at different scales; from molecular level to population level [14]. The rhythms reflect a collective activity of neurons interconnected via synaptic connections. Besides, neuron dynamics are determined by a specific balance of ionic currents [15]. These neuron intrinsic properties are controlled by neuroactive chemicals, called *neuromodulators*. Transition from sleep to wakefulness is associated with massive modifications in neuromodulators levels, such as serotonin, norepinephrine, dopamine and acetylcholine. These substances alter the behavior of thalamocortical neurons inducing shifts in population activity [5, 6, 9, 11, 13].

In parallel with these rhythm fluctuations, learning and memory are attributed to the ability of neurons to modify their connections with other cells based on experience, a property called *synaptic plasticity* [16, 17]. Synaptic plasticity mechanisms often exploit the level of correlation in the activity of connected neurons (for example in spike-time dependent plasticity), and can therefore be affected by abrupt changes in neuronal excitability. Strong constraints are exerted on models of plasticity because neural circuits are adaptable to help animals to modify their behavior. At the same time, these circuits must be stable in spite of these changes. There is a balance between adaptability and stability [18, 19].

The coexistence of these mechanisms raises challenging questions: how can switches in brain states remain reliable despite constant rewiring of neuron connectivity? How is synaptic plasticity affected by switches in brain states? Indeed, little is known about how shifts in network rhythms influence synaptic plasticity, hence learning. One reason for this puzzle is that state-of-the-art computational models of switches in brain states have often focused on the role of connectivity changes in network rhythm modulation [20–23]. Such models are not appropriate to study the impact of transient network oscillations on synaptic plasticity and learning, since the rhythmic switch itself relies on a disruption of the connectivity established through learning.

Recent evidence suggests that control of rhythmic activity can happen *at the cellular level*. No matter how fast or long-lasting are the transitions, they are controlled without affecting neither synaptic strength nor the circuit interconnection topology [24]. Such mechanisms

have been studied in small circuits, prominent examples being the crustacean stomatogastric system [24–28], and the leech heart [29, 30]. These circuits are ideal supports to better understand how alteration in excitability regulates behavioral states. These works demonstrate that neuromodulation generates highly stable outputs. It has also been extensively shown that several combinations of ionic channels on the membrane or synaptic connections lead to the desired outcome [27, 31]. It enhances the idea that rhythms cannot rely on a precise synaptic weight configuration or precise tuning of intrinsic parameters.

In this line of work; we aim at highlighting a cellular property that is critical for the generation of switches in brain states compatible with neuromodulation, cellular heterogeneity, synaptic plasticity and independent from network topology. This cellular property relies on a timescale separation between sodium and T-type calcium channel activations, providing a source of both fast and slow positive feedback loops at the cellular level. Slow positive feedback is accessible to all neurons that embed a sufficient amount of slowly activating voltage-gated calcium channels or slowly inactivating potassium channels in their membrane: the positive feedback comes from the fact that a calcium channel activation (resp. potassium channel inactivation) further amplifies the depolarization that gave rise to it, and slow means that calcium channel activation (resp. potassium channel inactivation) is at least one timescale slower than sodium channel activation. But the ultraslow inactivation of T-type calcium channels makes the slow positive feedback tunable: its gain depends on neuron polarization level. The presence of a slow positive feedback at the cellular level endows the neuron with an excitability switch, especially for the transition between tonic mode to bursting mode [32].

This present work studies the role played by the timescale separation between sodium and T-type calcium channel activations on the robustness of six published thalamic neuron models. The thalamic neurons raise interest due to their varying firing patterns and contribution to brain states. Two models among the analyzed in this paper neglect this timescale separation by designing T-type calcium channel activation as an instantaneous event, a simplification often encounters in neuronal modeling. Here, we show through several computational experiments that the compatibility between neuromodulation, synaptic plasticity and switches in brain states correlates with the presence of the *slow* T-type calcium channel activation. As soon as intrinsic parameters and synaptic weights are affected respectively by neuromodulation and synaptic plasticity, the two models that speed up the calcium activation kinetics experience a drastic drop in their switching capabilities. However, restoring the slow activation of T-type calcium channels in these two models improves the robustness of their rhythmic activity.

To further quantify the importance of a timescale separation between sodium and T-type calcium channel activations, we vary T-type calcium channel activation kinetics in all models, ranging from the fast timescale of sodium channel activation to the ultraslow timescale of T-type calcium channel inactivation, and we test its robustness at the circuit level and at the population level. Our computational experiments confirm that the robustness of rhythmic activity is achieved when T-type calcium channel activation is an order of magnitude slower than sodium channel activation. This was observed in all models despite their quantitative differences.

We also analyze this timescale separation from a dynamical system approach. We reduce the high-dimensional conductance-based models into models with three variables; fast ($V$), slow ($V_s$) and ultraslow ($V_u$). Each variable of the original model is decomposed into its contribution in $V$, $V_s$ and $V_u$ using the logarithmic distance proposed in [33], which permits to track the effect of changes in gating time-constants. The slow-fast phase portrait of models with slow T-type calcium channel activation are all qualitatively equivalent: the robust switch to bursting is generated by the appearance of a lower branch in the V-nullcline [32, 34–36]. Speeding-up

the activation of T-type calcium channels disrupts the appearance of this lower branch, which in turn disrupts the ability to switch to bursting.

Our results thus highlight the importance of respecting the physiological timescale separation between sodium and T-type calcium channel activations to guarantee compatibility between neuromodulation, synaptic plasticity, cellular heterogeneity, adaptable connectivity and switches in neuronal rhythms.

## Results

### Robust vs. fragile firing pattern transition at the single-cell level

Throughout this paper, we compared six well-established conductance-based models of thalamic neurons [37] (model 1), [38] (model 2), [39] (model 3), [40, 41] (model 4), [42] (model 5) and [43] (model 6). All these models include at least a sodium current, $I_{Na}$, a potassium current, $I_K$, a T-type calcium, $I_{CaT}$ and a leak current, $I_{leak}$ (see S1 Supplementary Material for more details). Each of these models is conceived to reproduce the different firing patterns observed in a thalamic neuron (a depolarized tonic mode, a hyperpolarized bursting mode, rebound bursting, etc.) and the switch between them [6, 7, 9, 44]. The firing mode is controlled by the external current. A depolarizing current drives the neuron model in tonic mode. If it is followed by a hyperpolarizing current, it switches the neuron model from a regular spiking mode to a bursting mode; a transition called hyperpolarized-induced bursting (HIB) (see Fig 1A) [6].

Experimental and computational studies have shown that a similar behavior or a similar firing pattern can emerge from neurons or circuits having very distinct intrinsic parameters [45–47]. Fig 1B illustrates the membrane voltage time-course of a thalamic neuron model for two different sets of maximal intrinsic conductances under the control of an external hyperpolarizing current (see black curves). The firing pattern is similar in both parameter sets and shows the typical switch occurring in thalamic neurons. However, the corresponding currentscapes reveal a variability in the contributions of the different ionic currents [45]. A H-current (orange curve) is involved in the first neuron (top currentscape) while it is almost absent in the second neuron (bottom currentscape). It shows that different combinations of ionic currents can lead to same firing pattern. This simple experiment motivated the rest of this work; a computational model must be able to reproduce a desired outcome for a broad range of intrinsic parameters as it happens in biology.

Here, we studied the robustness of conductance-based models to parameter variations with a special focus on the dynamics of the voltage-gated T-type calcium channel activation. The first four models (models 1 to 4) incorporate a *slow* activation of the T-type calcium (CaT) channels while the last two models (models 5 and 6) fix the activation as an instantaneous event. This simplification is often encountered in neuronal modeling [42, 43, 48–53]. Indeed, it removes one differential equation, which decreases the simulation time—a computation intake that is non-negligible as soon as one moves toward network simulations.

First, we investigated the impact of the CaT channel activation *dynamics* in the model robustness at the single cell level. To do so, we simply tested the ability to reproduce the switch from tonic to burst by changing a single parameter in the model, namely, the capacitance membrane $C_m$. An alteration in this parameter substitutes a change in cell size or a uniform scaling of all maximal ionic conductances [35, 54]. Fig 1C reveals the striking consequence for the model robustness when the capacitance is scaled by a factor 1/10. Models 1 to 4 (left panel) including the slow activation of T-type calcium channel are able to reproduce the hyperpolarized-induced bursting while models that assume this activation is instantaneous are fragile. Indeed, models 5 and 6 (center panel) loose the ability to switch from tonic to burst. To make the comparison as fair as possible, we have restored the slow dynamics of the CaT channel

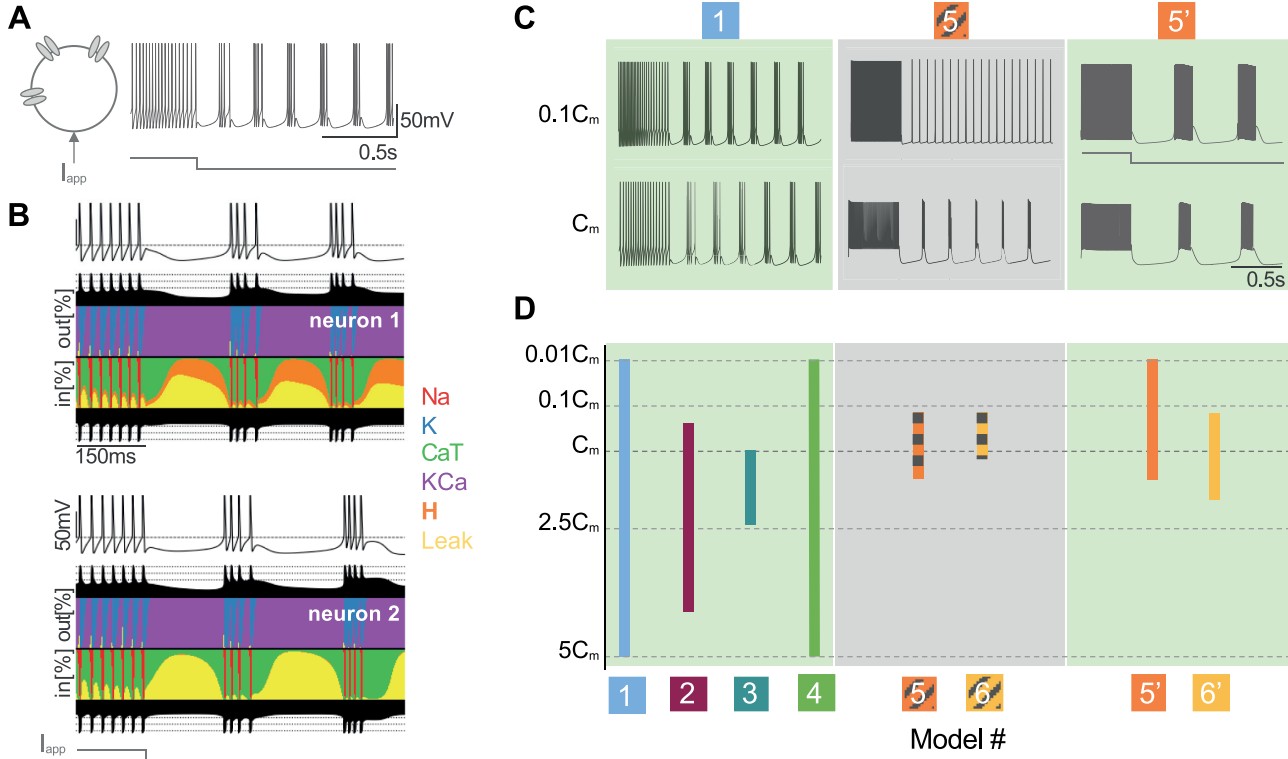

**Fig 1. Slow T-type calcium channel activation ensures model robustness against neuron variability at single-cell level. A**: The cell switches from a regular tonic mode to a bursting mode when a hyperpolarizing current is applied, a typical response called hyperpolarized-induced bursting (HIB). **B**: Currentscape for a neuron model with two different sets of maximal intrinsic conductances. After 150ms, the two neurons are hyperpolarized, leading to a HIB. Under the membrane-time curve, the black surface shows respectively on the top and bottom total inward outward currents on a logarithmic scale (see [45]). Between the black surfaces, each color curve reveals the contribution of one particular ionic current as the percentage of the total current during the simulation. Both neurons display the same firing pattern but this outcome is achieved by different combinations of ion channels densities. **C**: Models 1, 2, 3, 4, 5' and 6' are robust to a uniform scaling of all the maximal conductances, modeled by a change in membrane capacitance ($C_m$)(left panel). Models 5 and 6 are fragile to this parameter alteration. They lose the ability to switch (center panel). Restoring the slow CaT channel activation is enough to recover the robustness (right panel). **D**: Quantitative analysis of neuron model robustness to change in membrane capacitance. Each model is launched for a capacitance value varying from a hundredth to five times its nominal value. Models 1, 2, 3, 4, 5' and 6' cover larger parameter range. By contrast, models lacking of slow CaT channel activation are fragile to membrane capacitance deviation. Replacing the instantaneous CaT channel activation by a slow activation turns these fragile models into more robust models. Hence, models 5' and 6' support larger variation of $C_m$.

activation in models 5 and 6. We reconstructed a differential equation for the activation variable whose time constant is voltage-dependent. Models 5' and 6' are their respective modified versions. The T-type calcium current previously described by $I_{CaT} = g_{CaT} \boldsymbol{m_{CaT,\infty}^a} h_{CaT}(V_m - V_{Ca})$ is replaced by $I_{CaT} = g_{CaT} \boldsymbol{m_{CaT}^a}(V_m) h_{CaT}(V_m - V_{Ca})$ where $dm_{CaT}/dt = (m_{CaT,\infty} - m_{CaT})/\tau_{m_{CaT}}(V_m)$ (see S1 Supplementary Material for details). This only modification is sufficient to recover the desired firing activity, *ie*. the switch from tonic to burst even for a division of the capacitance membrane by a factor of 10 (Fig 1C, right panel).

This computational experiment has been reproduced for membrane capacitance values scaled from a hundredth to five times its nominal value (see Methods for details). Fig 1D reveals that models 1 to 4 are able to switch for a large range of capacitance values (see left panel). Models 5 and 6 are fragile and cover a tiny range around the nominal value ($C_m$) (center panel). Reinstating the slow dynamics of this channel bounces back the robustness as shown by the increase of the orange and yellow bars (right panel). Models 5' and 6' are able to generate the firing pattern switch typical in thalamic cells given capacitance values for which models 5 and 6 are not able.

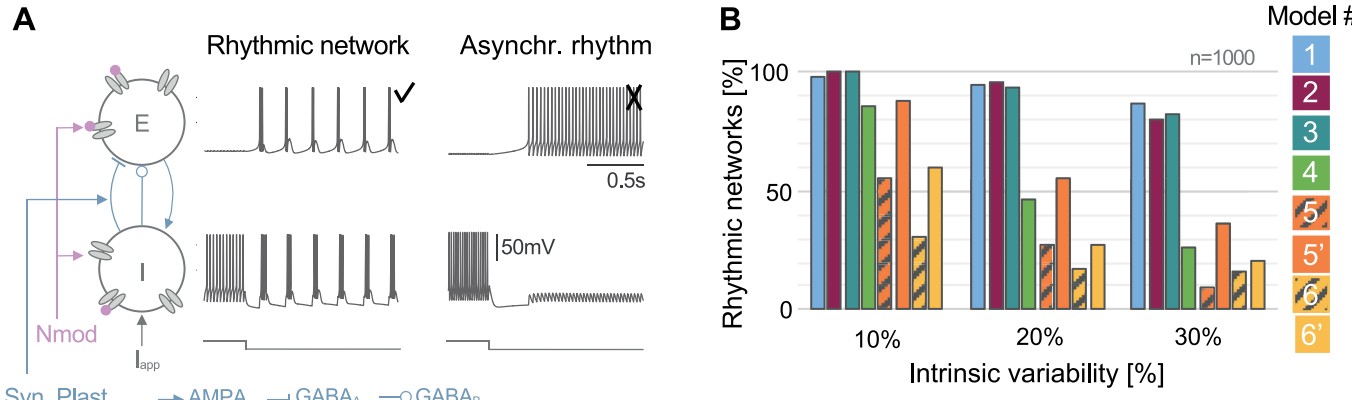

**Fig 2. Slow T-type calcium channel activation ensures model compatibility with neuromodulation and synaptic plasticity at the circuit level. A**: (left) Two interconnected neurons (one excitatory neuron E and one inhibitory neuron I) under the control of an external current ($I_{app}$), affected by neuromodulation (Nmod, pink spheres) and synaptic plasticity (Syn. Plast., in grayish blue). (center) The external current initially depolarizes the inhibitory cell then hyperpolarizes it leading to a switch in the circuit rhythm into a synchronous bursting. (right) Voltage traces illustrating an asynchronous rhythm, undesired behavior in the circuit. **B**: Percentage of rhythmic networks observed in different neuron models as intrinsic variability increases. For each model, one thousand 2-cell circuits are generated with random ionic conductances varying from 10, 20 and 30% from their nominal values -mimicking the effect of neuromodulation, and synaptic conductances, randomly picked in a uniform distribution—mimicking the effect of synaptic plasticity. Models 1 to 3 embedding a slow activation of T-type calcium channels are robust to parameter variability. Model 4 is robust at low variability than its performance is decreased due to its high number of ionic channels. Models 5 and 6 that assume an instantaneous T-type calcium channel activation are fragile: parameter variations disrupt the nominal rhythm. Replacing the instantaneous activation into a slow activation restores the robustness as shown by models 5' and 6' (dashed orange and yellow bars).

## Slow T-type calcium channel activation makes an isolated excitatory-inhibitory circuit robust to neuromodulation and synaptic plasticity

To extend our results obtained at the single-cell level, we moved to the circuit level. We built an isolated excitatory-inhibitory circuit of two neurons. These neurons are connected through AMPA, GABA$_A$ and GABA$_B$ connections to model the asymmetric coupling between a sub-population of excitatory (E) cells and a subpopulation of inhibitory (I) cells [4, 7]. This topology is a typical configuration in the thalamus [6, 8]. The E-I circuit and its expected rhythmic network activity are illustrated in Fig 2A (left and center panels). It is controlled by an external current injected on the inhibitory cell. Initially depolarized, the I-cell exhibits a tonic mode. The E-cell remains silent. As soon as the external current hyperpolarizes the I-cell; it deinactivates the T-type calcium channels, leading to a bursting mode. Then, thanks to the reciprocal connections, the circuit switches to a synchronous burst called the oscillatory mode.

With a simple computational experiment, we studied the robustness of these network switches to changes in neuron intrinsic properties, mimicking the effect of neuromodulation (modeled by maximal conductances), and changes in synaptic weights, mimicking the effect of synaptic plasticity (modeled by the synaptic conductances). For each model, we started from an E-I circuit capable of generating a switch. Then, one thousand 2-cell circuits were simulated for different parameter sets of maximal intrinsic conductances and synaptic weights. The maximal conductances varied within an interval of 10, 20 and 30% around their nominal values and the synaptic weights varied in a fixed range (see Methods for details). The percentage among the thousand 2-cell circuits that have performed the rhythmic transition was evaluated for the three intervals of variability in each conductance-based model. A rhythmic network is defined according to the firing pattern evolution shown in Fig 2A (center) representing a switch from silent-tonic to synchronous bursting while other activities are classified as an asynchronous rhythm, for example the pattern in Fig 2A (right). Quantification of the firing pattern properties *i.e.* frequencies in tonic and burst is available in S1 Supplementary Material.

For a small intrinsic variability (10%), Fig 2B shows that models 1 to 4 are robust to neuromodulation and synaptic plasticity. More than 800 sets of parameters allow the circuit to switch. The absence of slow positive feedback in models 5 and 6 has a dramatic consequence on the model robustness. One every two parameter sets in model 5 cannot reproduce the typical thalamic activity transition. Model 6 is even more fragile. However, restoring the dynamical cellular property significantly improves the robustness as shown in models 5' and 6'.

For larger intrinsic variability (20% and 30%), models 1 to 3 maintain their capabilities to switch for a broad range of intrinsic and synaptic parameter ranges. Model 4 can be considered apart. It is shrinking as the variability increases. This decrease in performance is likely related to its high number of conductances (about twice as much as the other models). Indeed, we are exploring a 14-dimension space since this model has 11 intrinsic conductances and 3 synaptic conductances. The parameter exploration for the other models occurs in a 7 to 9 dimension space reducing the model complexity. Models 5 and 6 have a similar number of intrinsic conductances as models 1 to 3 but they are extremely fragile to parameter changes. They are almost unable to perform rhythmic transition. The rhythm in an E-I circuit requires a precise tuning of intrinsic and synaptic parameters for models lacking of the slow kinetics of CaT channel activation. This computational choice makes the model incompatible with neuromodulation and synaptic plasticity. Once again, their modified versions embedding the slow positive feedback (correlated to the slow activation of T-type calcium channels) have a better response to parameter perturbations.

## A timescale separation between sodium and T-type calcium channel activations ensures compatibility between circuit switch, neuromodulation and synaptic plasticity

So far, we have shown that modeling the T-type calcium channel activation with a slow kinetics drastically enhances the robustness of rhythmic switches at the cellular, and circuit levels. But one question remains: what does *slow* mean, and how tuned does the activation kinetics need to be to achieve robustness? Indeed, there exists different subtypes of T-type calcium channels whose activation kinetics can greatly differ [55, 56].

To answer this question, we explored the impact of incrementally varying T-type calcium channel activation kinetics in a similar computational experiment as done in Fig 2. We focused on models 1, 2, 3, 4, 5' and 6' that describe the opening of the T-type calcium channel with a first order differential equation $dm_{CaT}(V_m)/dt = (m_{CaT,\infty} - m_{CaT})/\tau_{m_{CaT}}(V_m)$. The variable $\tau_{m_{CaT}}$ is the voltage-dependent activation time constant and characterizes the dynamics of channel opening. We started from an isolated 2-cell E-I circuit connected via AMPA, GABA$_A$ and GABA$_B$ synapses. The circuit is able to switch from tonic to burst at the nominal parameter values. Then, we added neuromodulatory effect—by varying intrinsic parameters, and synaptic plasticity—by changing extrinsic parameters. We built four hundred 2-cell circuits whose maximal ionic conductances and synaptic conductances were randomly picked in an interval of 20% around their basal values. Here, the novelty was to play with the time constant of CaT channel activation $\tau_{m_{CaT}}$ of the two neurons (see Fig 3A).

Fig 3B is a comparative table between time constants associated to sodium channel activation $\tau_{m_{Na}}$, T-type calcium channel activation $\tau_{m_{CaT}}$ and inactivation $\tau_{h_{CaT}}$ evaluated at their threshold voltage (see Methods for details). It points out the quantitative differences between models. Here, we were investigating the choice made for $\tau_{m_{CaT}}$ with respect to $\tau_{m_{Na}}$ and $\tau_{h_{CaT}}$. To do so, the time constant $\tau_{m_{CaT}}$ was scaled by several multiplicative factors from 0.01 to 100 times its nominal value. The smallest the coefficient, the fastest the CaT channel activation.

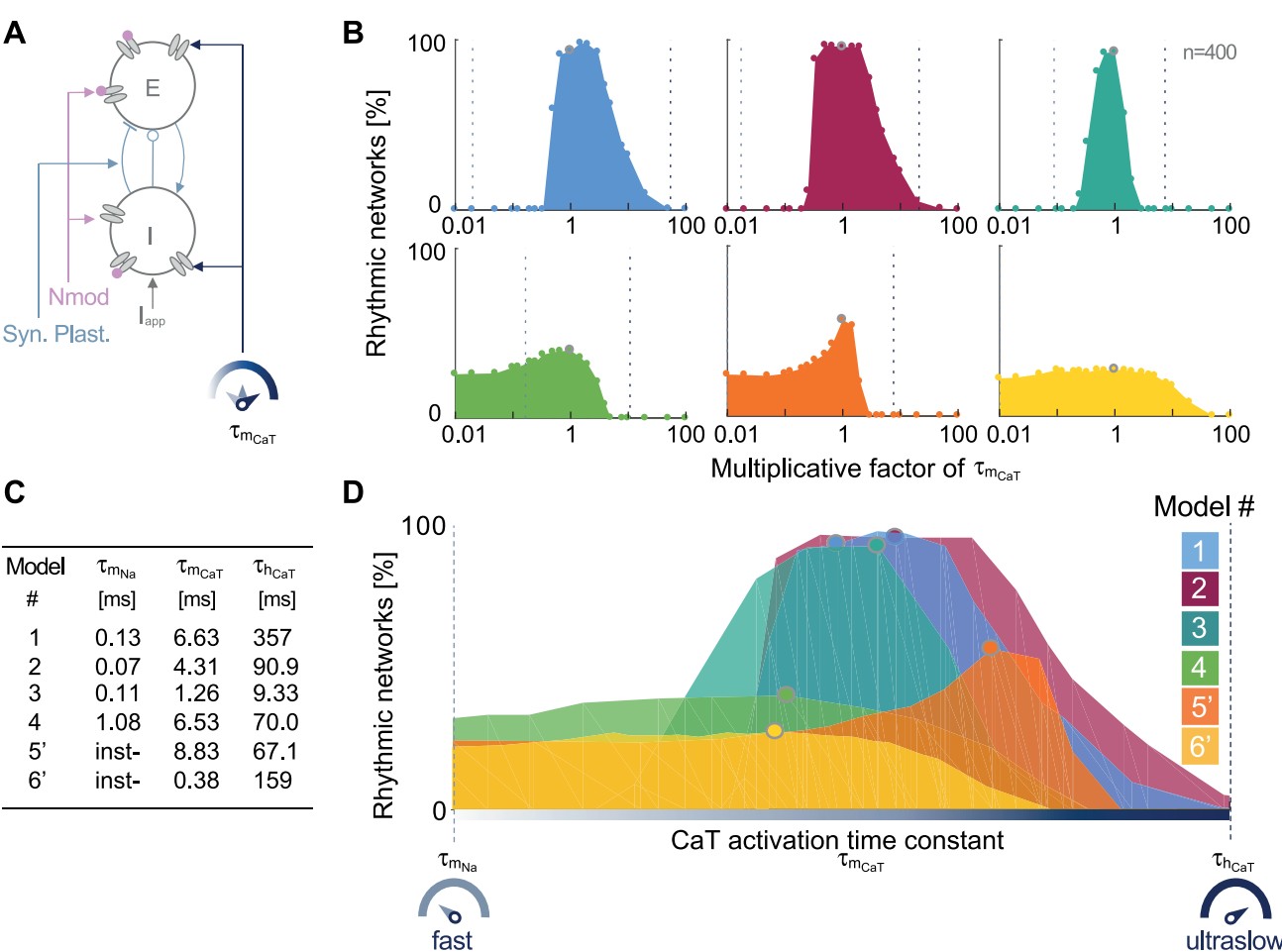

**Fig 3. The physiological *slow* timescale of T-type calcium channel activation guarantees compatibility between circuit switch, neuromodulation and synaptic plasticity. A**: For each model, 400 E-I 2-cell circuits are built from scaled intrinsic and synaptic conductances picked in a uniform range of ± 20% around their nominal values to mimic neuromodulation and synaptic plasticity. These same 400 circuits are then simulated for a varying CaT channel activation time constant ($\tau_{m_{CaT}}$). **B**: Comparison table between the time constants associated with sodium channel activation ($\tau_{m_{Na}}$), CaT channel activation and CaT channel inactivation ($\tau_{h_{CaT}}$). **C**: Effect of a varying CaT channel activation time constant on the switching capability in a 2-cell circuit for models 1, 2, 3, 4, 5' and 6'. The y-axis quantifies the percent of rhythmic networks among the 400 simulated random circuits under different values of $\tau_{m_{CaT}}$, scaled by a multiplicative factor varying from 0.01 to 100. The performance associated with the nominal (resp. scaled) CaT channel activation is depicted with a gray circle (resp full circle). The sodium channel activation (resp. CaT channel inactivation) time constant is marked with the left (resp. right) dashed vertical line. **D**: Each model is superposed between a window bounded on the left (resp. right) by the sodium channel activation (resp. CaT channel inactivation) time constant. The CaT activation time constant must remain confined in the *slow* timescale in order to guarantee model robustness to neuromodulation and synaptic plasticity.

For each scaled CaT time constant, we tested the model capability to switch from tonic to burst when the I-cell is hyperpolarized. Among the 400 tested circuits, the percentage of rhythmic circuits is placed on the y-axis (see Fig 3C). The x-axis is on a logarithmic graduation.

When the multiplicative factor is equal to one (marked by the gray circle), it indicates the CaT channel activation time constant initially designed for each model. The sodium channel activation time constant $\tau_{m_{Na}}$ and the CaT channel inactivation time constant $\tau_{h_{CaT}}$ are also drawn for each model in dashed lines (respectively $\tau_{m_{Na}}$ on the left and $\tau_{h_{CaT}}$ on the right). They are evaluated at their threshold voltage (see Methods for details). They respectively indicate the *fast* and the *ultraslow* timescales. For models 5' and 6', the sodium current activation is

instantaneous. For model 6', the CaT channel inactivation does not appear on the graph since it is greater than 100 times its activation (see Table in Fig 3B).

The outcome of this computational experiment is compelling. Fig 3C reveals that models are robust to neuromodulation and synaptic plasticity when the timescale of the CaT channel activation is situated in a slow range. The meaning of slow stands by itself; it is bounded between the fast kinetics of the sodium channel activation and the ultraslow kinetics of CaT channel inactivation. For each model, the peak of robustness lies between these two timescales. The bump-shaped surface shows a relatively large width. It points out that the activation kinetics do not need to be perfectly equal to one specific value between the fast and ultraslow regions. But these kinetics just need to be included within the slow timescale range. However, as soon as the kinetics moves too far from this interval, the robustness loss is abrupt.

To go further in the analysis, the six models were superposed on each other by normalizing the logarithmic x-axis on Fig 3D. The left (resp. right) boundary is the time constant of the sodium channel activation (resp. CaT channel inactivation); namely, the fast and the ultraslow timescales. The number of rhythmic circuits is enhanced when the CaT activation occurs at a timescale slower than the sodium activation and faster than the CaT inactivation as highlighted by the bump-shaped surface. Modeling the CaT channel opening at a slow timescale guarantees the compatibility between neuromodulation, synaptic plasticity and switches in brain states. Indeed, the compatibility relies on the presence of a *slow positive feedback* at the cellular level, as mentioned above.

Model 4 maintains a steady robustness even if the CaT channel activation is accelerated. This can be explained by the presence of another source of slow positive feedback such as a slowly activating L-type calcium channels (see S1 Supplementary Material for more details about model 4). Models 5' and 6' display a modest robustness for a fast opening of the CaT channel. These models were initially designed to operate for an instantaneous activation. However, the favorable operating point is preferably at a slow timescale.

If the kinetics of the CaT channel opening slows down too much (meaning we are moving to the right on the x-axis), it reaches the same timescale as its inactivation. In other words, the activation gate opens while the inactivation gate closes leading to a zero flux of calcium ions. The kinetics of CaT channel activation must be slow but not too slow.

## A timescale separation between sodium and T-type calcium channel activations promotes robustness of network states in large heterogeneous populations

From an isolated E-I circuit of two neurons, we built a larger network whose topology is emblematic of the thalamus, illustrating the interaction between relay neurons and the reticular nucleus. This population interaction is involved in state regulation such as the transition from wakefulness to sleep [6, 9]. We replicated the two previous computational experiments performed at the circuit level, now on a neuronal population. To do so, we started with a 200-cell network where the population of 100 excitatory neurons is identical to the population of 100 inhibitory neurons. We neglected intra-population interaction, and we assumed all-to-all connectivity between the two populations. The E-cells projected AMPA synapses to all the I-cells and conversely, the I-cells were connected to the E-cells via $GABA_A$ and $GABA_B$ synapses. All the synaptic weights linking the neurons together were identical. An external current was exerted on the inhibitory cells (see Fig 4A). Hyperpolarizing this current caused a cellular switch and drives the neurons in a synchronous bursting mode as previously shown with the isolated E-I circuit in Figs 2A and 4B (voltage traces, two top curves) [37]. This change in cellular firing pattern is translated by an oscillatory behavior at the network level. This oscillatory

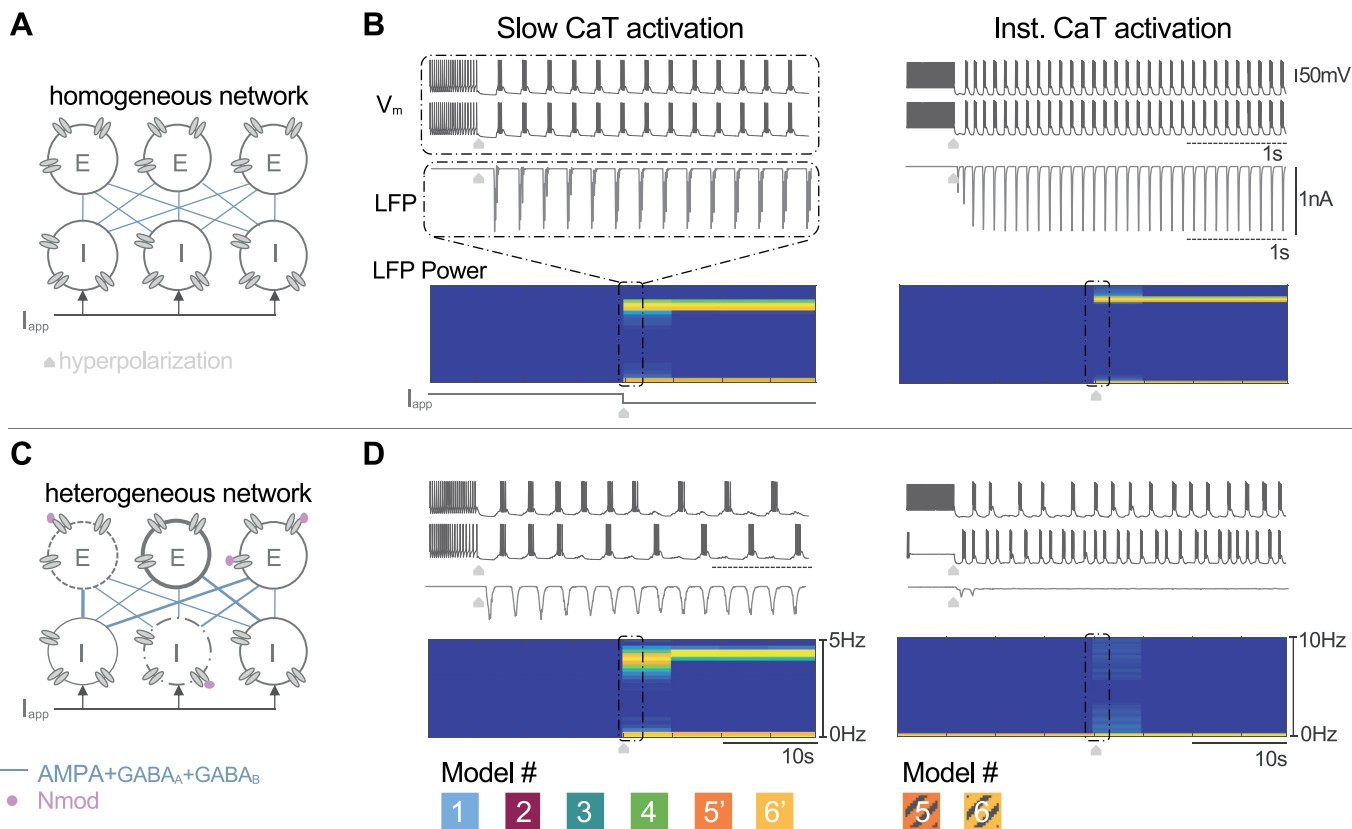

**Fig 4. Slow T-type calcium channel activation guarantees robust network switch independent on the population heterogeneity. A**: A 200-cell network is built with the same neuron model; 100 excitatory cells connected via the AMPA synapses to 100 inhibitory cells projecting back GABA$_A$ and GABA$_B$ synapses. The network is homogeneous. All neurons have the same channel densities and the same synaptic weights. **B**: Voltage traces of two inhibitory cells (two top curves), time-course (third curve) and frequency-time graph (bottom spectrogram) of the local field potentials (LFPs) of the inhibitory neuron population for the homogeneous network. Hyperpolarization of the inhibitory neurons turns on the mean-field rhythm activity of the population depicted by a synchronous bursting at the cellular level, a oscillating synaptic activity shown on the LFP time course whose frequency is shown by the high power LFP frequency band on the spectrogram. **C**: A heterogeneous 200-cell network is built to take into account neuromodulation, cell variability, more representative topology and synaptic plasticity. Each ionic and synaptic conductance is randomly picked in a given range around its nominal value (see Methods for details). **D**: (left panel) Only models 1,2,3,4 5' and 6' display the switch in the mean field rhythm of the population marked with an oscillatory LFP time-course (third curve) and a significant power band in the spectrogram (bottom). (right panel) Models 5 and 6 that lack slow CaT channel activation are fragile to the network topology and the heterogeneity. Switch in population rhythm is recovered when the slow regenerativity is restored (models 5' and 6').

state can be visualized by computing the local field potential (LFP) of the neuron population. LFP is measured as the sum of synaptic activity in a neuronal population (see Fig 4B, third curve). When the current hyperpolarizes the I-cells, the synaptic current is remarkably modified and reveals a stronger activity. The spectrogram of the LFP shows that the hyperpolarizing current turns on the mean-field rhythmic activity marked by a strong power LFP frequency band (see Fig 4B, frequency-time image at the bottom). For each model, the homogeneous network is able to switch from an active state to an oscillatory state.

However, a perfect network with identical neurons and identical synaptic weights is not consistent with reality. Each neuron differs from its neighbor with different intrinsic parameters such as the cell size or the densities of ionic channels. In addition, connections between neurons are neither identical nor static. Therefore, we explored the model ability to maintain switch in brain states in presence of cellular heterogeneity and its independence on network topology [37, 57]. To do so, we built a 200-cell network where each neuron is different and the connectivity is uneven (see Fig 4C). The intrinsic and synaptic parameters were randomly

picked in a given interval (see Methods for details). Fig 4D shows the astonishing contrast between the stability of models including the slow CaT channel activation (see left panel) and the fragility of models lacking of this property (see right panel). Models 1 to 4, 5' and 6' are still able to generate a switch into a synchronous burst as shown in Fig 4D (left). Due to intrinsic and synaptic variabilities, the voltage recordings from two I-cells are more realistic (two top traces). Even if each cell is not perfectly bursting at each cycle, the summation of the synaptic activity shown by the LFP curve is oscillating(third curve). The frequency of this oscillation is quantified in the spectrogram (bottom frequency-time graph). Models 5 and 6 are able to switch from an active to an oscillatory state when the network is homogeneous and the connectivity is perfectly balanced. However, as soon as the network is changed into a more realistic configuration, these models cannot preserve switches in brain states as shown by the asynchronous voltage-traces and the flat LFP curve translated by the absence of a marked power band in the spectrogram (see Fig 4D right panels).

Models 1 to 3 show a marked power band in their spectrogram when intrinsic parameters vary in an interval of 20% around their nominal values. Once again, model 4 is less robust, certainly due to its high number of conductances. It continues to switch for an intrinsic variability of 10%. Model 5 (resp. model 6) does not tolerate a variability of 20% (resp. 5%). When the slow activation of T-type calcium channels is reestablished, models 5' and 6' switch from an active state to an oscillatory state at the same level of variability that the initial model was fragile. This modeling modification leads to a neuron model robust to cell variability and that does not rely on the network topology.

To go further, we explored once again the relevance of respecting physiological timescale separation in ionic current modeling but this time at the population level. To do so, we built a 200-cell network with 100 excitatory cells connected to 100 inhibitory cells via AMPA, $GABA_A$ and $GABA_B$ synapses. Models 1,2, 3, 4, 5' and 6' were switching their network state under a hyperpolarizing current for a homogeneous and a heterogeneous configuration, as shown in Fig 4B and 4D (left panels). Here, we tested which kinetics provides robustness to network heterogeneity. The CaT activation time constant $\tau_{m_{CaT}}$ of the 200 neurons was scaled by several multiplicative factors ranging from an eighth to eight times its nominal value. In addition, cellular and synaptic variabilities were introduced by randomly picking the maximal intrinsic and extrinsic conductances of each neuron in a given interval around their nominal values following a uniform distribution (see Fig 5A). These heterogeneous networks were constructed with parameters varying in an interval whose width was ranging from 0 to 50% with a step of 5%. At each scaled $\tau_{m_{CaT}}$, we tested the maximal possible variability width at which the heterogeneous network is able to switch. For example, we built the 200-cell network whose maximal ionic and synaptic conductances were picked in a range of 20% around their initial values. Then, we simulated the network under the control of an hyperpolarizing current at different scaled $\tau_{m_{CaT}}$. We finally checked for each timescale if the given heterogeneous network was switching or not, by analyzing its LFP activity. The network was switching if the time course of neuron population LFP displayed an oscillatory behavior in the hyperpolarized state and if its spectrogram was marked by a strong power band (see Fig 5B). We continued to increase the variability interval to quantify the correlation between the timescale and its robustness to cellular heterogeneity and uneven topology.

Fig 5C summarizes the model robustness at each scaled CaT time constant. It confirms our previous result; accelerating or decelerating the CaT channel opening makes the six models fragile to heterogeneity. In models 1 to 3, the best operating point to set the CaT time constant is confined between the fast timescale and the ultraslow timescale as shown by the darker zone. It enhances the model capability to switch in presence of cellular heterogeneity and

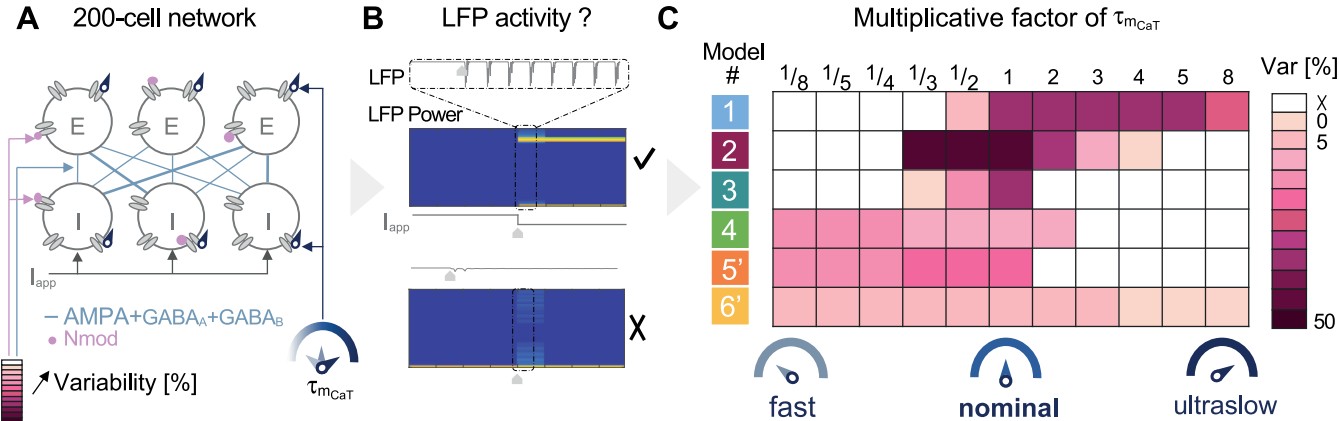

**Fig 5. Comparison between fast, slow or ultraslow T-type calcium channel activation in generating robust mean-field activity transition. A**: A 200-cell network with 100 excitatory neurons and 100 inhibitory neurons connected via AMPA, GABA$_A$ and GABA$_B$ synapses under the control of an hyperpolarizing current. The intrinsic and extrinsic parameters are respectively affected by neuromodulation and synaptic plasticity; their values are randomly picked in a given interval (namely variability). The CaT channel activation time constant $\tau_{m_{CaT}}$ of each neuron is scaled by a multiplicative factor. **B**: At each scaled time constant, we check if the heterogeneous network is switching by analyzing the LFP activity. If the LFP timecourse presents a strong activity and its spectrogram shows a marked power band, the network displays an oscillatory state during the hyperpolarization state. **C**: The table summarizes the largest variability width at which the network presents a switch in its mean-field activity for several scaled time constants. Respecting the slow timescale of the CaT channel activation guarantees the switch in network rhythm compatible with variability in channel densities and synaptic weights. Driving the CaT channel activation to a fast or ultraslow timescale makes models more fragile to network topology and heterogeneity.

synaptic plasticity. Model 4 is not really robust due to its high number of ionic currents. It is also assumed to embed another source of slow regenerativity helping it to operate at a faster timescale. As exhibited in Fig 3C for an isolated 2-cell circuit, models 5' and 6' also maintain a certain ability to switch even at a faster timescale because they were initially designed to operate at an instantaneous opening of CaT channels. As adapted versions of models designed to switch with an instantaneous T-type calcium activation, their robustness is lower than models 1 to 3 in all parameter ranges.

Overall, Figs 3 and 5 reveal the importance of considering the physiological kinetics range of ion channel gating in computational models and especially the timescale separation between the different ionic currents. Getting rid of the slow dynamics of the T-type calcium channel activation disrupts the timescale separation with sodium channel activation. It removes an important biological property of this neuron type and thus, disturb its ability to change its firing pattern under a hyperpolarization in presence of neuromodulation and synaptic plasticity. This deficiency is transposed at the population level as shown by the inability to turn on the mean-field activity as soon as the cellular heterogeneity and unbalanced connectivity are increased.

## Slow T-type calcium channel shapes a robust phase portrait

How does the kinetics of T-type calcium channel shapes the phase portrait? To answer this question, we reduced the different high-dimensional models following the protocol presented in [33]. The protocol is based on the decomposition of each variable into their role in a fast, slow and ultraslow timescale using a logarithmic distance [33]. It allows us to reduce the different conductance-based models in a systematic and rigorous manner in order to obtain three dimensional models with three variables: the membrane voltage of the reduced model $V$, the slow variable $V_s$ and the ultraslow variable $V_u$. The contribution of the activation or inactivation channel variables ($m_i(V)$ or $h_i(V)$ contracted as $X(V)$ to ease the reading) is projected on these three variables. The voltage-dependent gating variable is transformed into a weighted

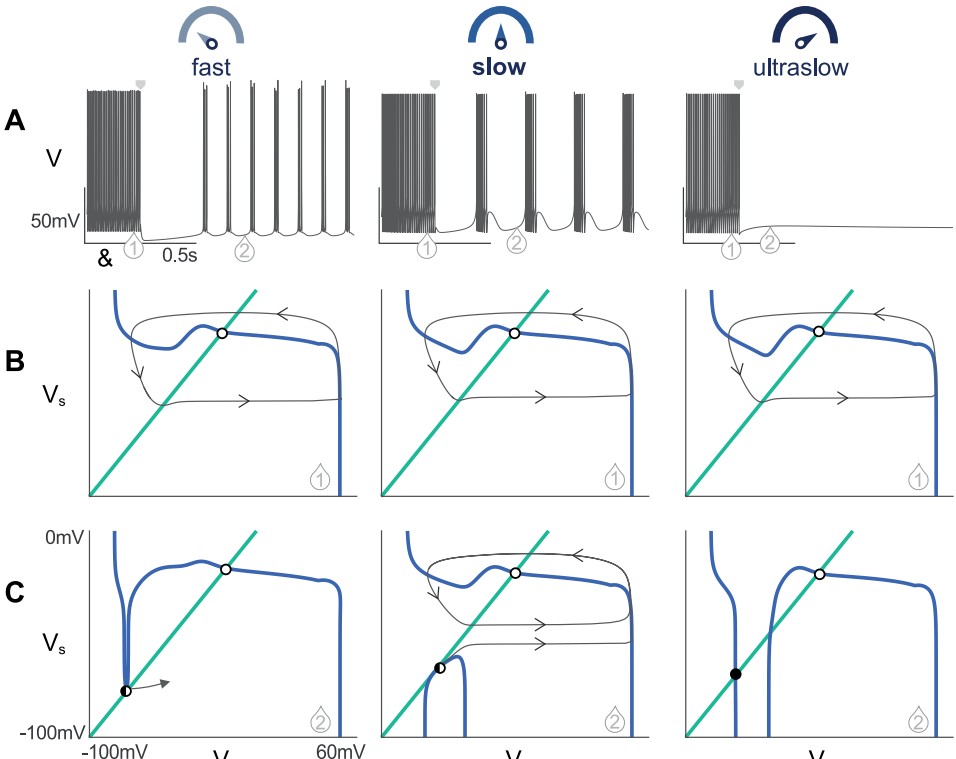

**Fig 6. Distortion of the phase portrait when the T-type calcium channel activation is considered as fast, slow or ultraslow. A**: Voltage traces of a reduced model under the three conditions during a hyperpolarized-induced bursting. The square arrow indicates the hyperpolarizing current step. When the CaT activation is ultraslow, the model is not able to switch from tonic to burst. **B**: Comparison of the portrait geometry during tonic mode (arrow 1) under the three timescales. V-(resp. Vs-) nullcline is sketched in blue (resp. green). The unstable fixed point is marked in open circle. The limit cycle followed by the trajectory is sketched in gray. The timescale chosen for the CaT channel activation is not affecting the tonic mode. **C**: Comparison of the portrait geometry during burst mode (at the saddle node bifurcation, arrow 2) under the three timescales. The stable fixed point is marked as a filled black circle and the saddle point meeting the stable fixed point is indicated by the black half-circle. At the ultraslow timescale (right), there is no saddle-node bifurcation, the V-nullcline is distorted into its hourglass shape trapping the trajectory into its stable fixed point. (left) For the *fast* CaT activation channel, the V-nullcline presents only the upper branch where the saddle node bifurcation occurs (center) For the *slow* activation, the V-nullcline exhibits a lower branch. This branch robustly separates the silent region and the spiking region of bursting. Videos of the simulations under the three conditions are available in S1 Video (fast), S2 Video (slow) and S3 Video (ultraslow).

sum of three terms associated with the three timescales where the weights are voltage-dependent. The weighted sum is written as: $X(V) = w_{fs}^X(V)X_\infty(V) + (w_{su}^X(V) - w_{fs}^X(V))X_\infty(V_s) + (1 - w_{su}^X(V))X_\infty(V_u)$ where $w_{fs}^X(V)$ is the contribution of the gating variable on the fast time-scale, $(w_{su}^X(V) - w_{fs}^X(V))$ is the contribution on the slow time-scale and $(1 - w_{su}^X(V))$ is the contribution on the ultraslow time-scale. Therefore, the 3D reduced model is described by the following differential equations: $C_m \, dV/dt = -\Sigma I_{ion} + I_{app}$, $dV_s/dt = (V - V_s)/\tau_{m_K}(V)$ and $dV_u/dt = (V - V_u)/\tau_{h_{CaT}}(V)$. The time-constant of the sodium activation $\tau_{m_{Na}}(V)$ paces the fast time-scale, the time-constant of the potassium activation $\tau_{m_K}(V)$ paces the slow time-scale and the inactivation of the T-type calcium channel $\tau_{h_{CaT}}(V)$ paces the ultraslow time-scale.

In this work, we focus on the timescale attributed to the activation to the T-type calcium channel. We investigated the distortion of the phase portrait geometry when this time constant is increased or decreased. Fig 6 shows the analysis of a conductance-based model reduced under three different conditions applied on T-type calcium channel activation, when it is

considered as fast, $(\tau_{m_{CaT}}(V)/50$—left column), slow (nominal $\tau_{m_{CaT}}(V)$—center column) and ultraslow ($50\tau_{m_{CaT}}(V)$—right column). Fig 6A shows the voltage time-courses when a hyperpolarizing current is applied to reproduce a hyperpolarized-induced bursting. Fig 6B exhibits the fast-slow phase portrait drawn at a given time (indicated by (1) in the voltage trace) in order to compare the tonic mode under the three conditions. Fig 6C is also the fast-slow phase portrait drawn this time at the saddle node bifurcation (indicated by 2 in the voltage trace, see Methods for details). Videos of the membrane voltage time course, the associated phase portrait including the evolution of the nullclines and the trajectory for the different kinetics are available in S1 Video (fast), S2 Video (slow) and S3 Video (ultraslow). In S1 Supplementary Material, simulations of the reduced models 2, 5', 6 and 6' are available.

First, the timescale chosen for CaT channel activation has no influence on the tonic mode. The reduced model is similarly spiking under the three conditions. The phase plane associated to the discharge mode is not affected. It shows the classical limit cycle extensively studied in spiking models. The trajectory is trapped in the limit cycle around the unstable fixed point present on the expected N-shaped V-nullcline. Second, considering the T-type calcium channel activation as slow as its inactivation removes the ability of the model to switch from tonic to burst (third column). The V-nullcline has a hourglass shape showing the influence of the inactivation of the T-type calcium channel (as shown in [58]). The trajectory is attracted by the stable fixed point at the hyperpolarized state. This highlights a lack of depolarizing current, which is due to the simultaneous activation and inactivation of T-type calcium channels.

Finally, the most interesting result is the comparison of the phase portraits during burst mode in the case of the fast activation (Fig 6C-left) versus the slow activation of T-type calcium channel (Fig 6C-center). We drawn both reduced models at the saddle node (SN) bifurcation (see Methods for details). The main difference comes from the V-nullcline: for the fast activation of CaT channel, the phase portrait is qualitatively similar to the one in spiking mode. For the nominal activation timescale, the phase portrait qualitatively changes by the appearance of a lower branch in the V-nullcline, which permits to robustly separate the silent region (which sits on the lower branch) and the spiking region of bursting (which sits on the upper branch).

Therefore, speeding up the activation of the T-type calcium activation disrupts the ability of T-type calcium channel deinactivation to qualitatively change the phase portrait structure from robust spiking to robust bursting in response to hyperpolarization [32, 34, 58]. The same behavior is observed in all models regardless of their quantitative differences. This remarkable distortion of the phase portrait when $\tau_{m_{CaT}}$ is scaled from the fast timescale to the ultraslow is shown in S4 Video.

The consequence on robustness and tunability of this qualitative change is illustrated in Fig 7. This figure compares the phase portrait of the reduced version of model 1(embedding a slow T-type calcium channel activation) and the reduced version of model 5 (embedding an instantaneous T-type calcium channel activation) when the capacitance membrane is divided by three. We have chosen the membrane capacitance for two reasons. First, as done in Fig 1 it is a suitable parameter to test the model robustness to a uniform scaling of maximal conductances or mimicking a change in cell size. Then, from a dynamical viewpoint, the membrane capacitance sets the timescale of the voltage equation. Reducing its value does not affect the nullcline shape nor the fixed point locations but solely affects the vector field.

Both models are able to switch from tonic to burst at the nominal value $C_m$, but they do so through two different mechanisms. Model 1 uses the appearance of a lower branch in the V-nullcline (see Fig 7A, left), whereas model 5 relies on the presence of a region where the timescale separation between the fast and slow timescales is inverted, *i.e.* the slow timescale becomes faster, which is illustrated by vertical trajectories and vector field away from the

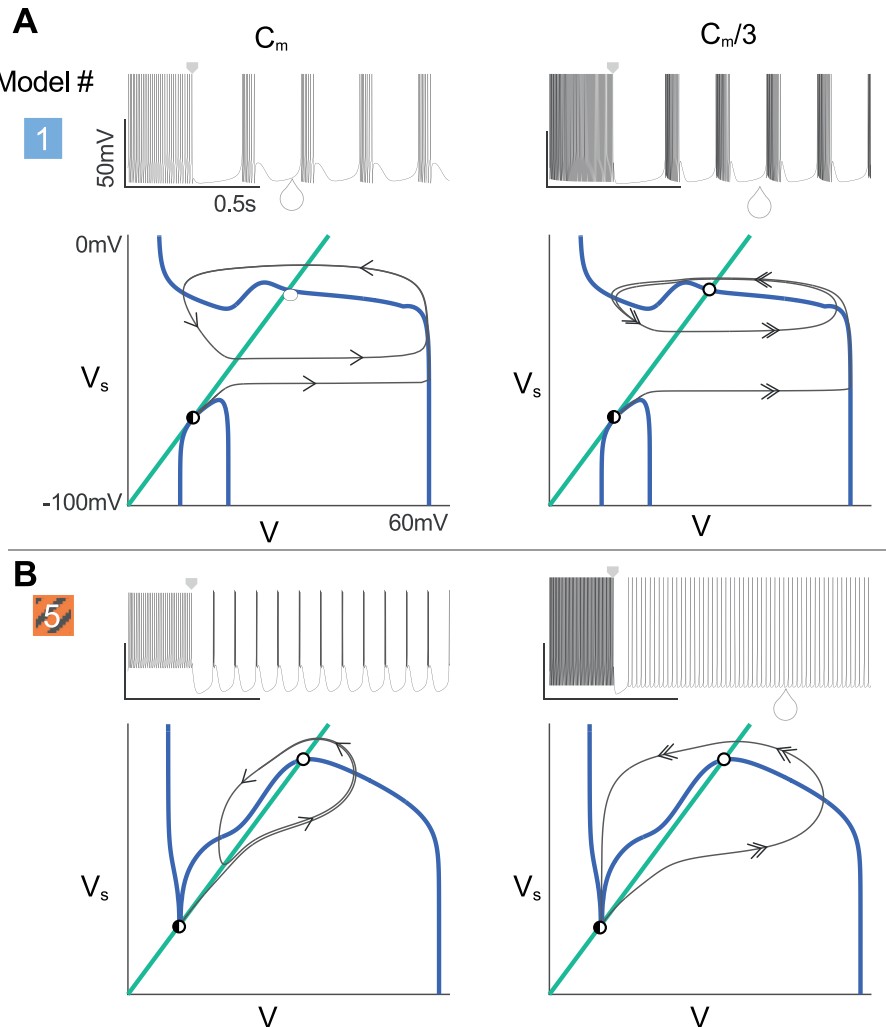

**Fig 7. The slow activation of T-type calcium channel reveals the appearance of a lower branch in the V-nullcline providing robustness to parameter variation. A-B:** (top) Recording of the membrane voltage during a hyperpolarized-induced bursting (bottom). Phase portrait at the saddle node bifurcation. V-(resp. Vs-) nullcline is sketched in blue (resp. green). The square arrow indicates the hyperpolarizing current step. The unstable fixed point is marked by an open circle and the saddle point meeting the stable fixed point is indicated by the black half-circle. The trajectory is sketched in gray and the arrow indicates the speed along the x-axis. (left to right) the membrane capacitance is divided by 3. The velocity along the x-axis is increased. **A:** The V-nullcline exhibits a lower branch making the phase portrait robust to membrane capacitance variation due to the sharp separation between the limit cycle and the hyperpolarized state. **B:** When the membrane capacitance is divided by three, the reduced model 5 is no more able to switch from tonic to burst. Small deviations destroy the rest-spike bistability. Videos of the different simulations are available in S5 Video to S8 Video.

nullclines (see Fig 7B, left). Both mechanisms permit to generate bursting. However, the second mechanism relying on such a dynamical property, it is very fragile to changes in membrane parameters: a reduction of the membrane capacitance removes this region of inverted time-scale separation, hence disrupts the ability to burst (Fig 7B, right). The mechanism based on the lower branch of the V-nullcline is structural, hence robust to changes in dynamics created by changes in membrane properties (Fig 7A, right). Videos of the simulations are available in S5 Video (model 1, $C_m$), S6 Video (model 1, $C_m/3$), S7 Video (model 5, $C_m$) and S8 Video (model 5, $C_m/3$).

The reader is referred to [32, 58, 59] for a detailed explanation of the origin and the effect of the lower branch present in the V-nullcline in model with the slow CaT calcium channel activation. More explanations about the robustness and tunability provided by the presence of the lower branch in the V-nullcline versus the classical N-shape are presented in [35].

## Discussion

### The physiological timescale separation between sodium and T-type calcium channel activations

A quotation from Bertil Hille's book [55], "The time course of rapid activation and inactivation of T-type $I_{Ca}$ has been described by an $m^3h$ model like that for $I_{Na}$ (Coulter et al.1989 [60]; Herrington and Lingle 1992 [61]), but the derived time constants $\tau_m$ and $\tau_h$ are 20 to 50 times longer than those for $I_{Na}$ of an axon at the same temperature.", highlights a physiological timescale separation between sodium and T-type calcium channel gating kinetics. T-type calcium channel presents of portfolio of activation kinetics depending on their isotype (i.e. $Ca_V3.1$, $Ca_V3.2$ or $Ca_V3.3$), but all isotypes share the property of activating a timescale slower than sodium channels: the activation time constant ranges between 1ms and 50ms [56, 62–67]. By contrast, the activation time constant of sodium channel is lower than 1ms [55, 68–70]. As explicitly shown with these numbers, there is one order of magnitude between these two gating time constants.

T-type calcium channels play a major role in oscillations generated, sustained or propagated by the thalamic circuit such as in sleep or in epilepsy. They are known to be modulated by several signaling cascades or also targets of several experimental and clinical drugs [56, 71–76]. These modulations can change the kinetics of this channel activation. Our computational experiments are physiologically relevant since modifying the activation kinetics can lead to undesired behavior.

### Modeling T-type calcium channel activation in conductance-based models

In a conductance-based model, the opening of an ion channel is represented by a voltage-dependent gating variable. This variable is described with a first order differential equation whose time constant is also voltage-dependent. In this paper, we investigated the kinetics chosen for the T-type calcium channel activation with respect to sodium channel activation. In other words, we compared these time constants with respect to each other.

There is a growing trend to design T-type calcium channel activation as the equivalent of a sodium channel activation. It often based on the fact that these two channels activate on a faster timescale than their own inactivation. For example in the published paper associated to model 5, it is said that "the activation variable [of T-type calcium channel] $s$ is relatively fast and is replaced by its equilibrium function. (. . .) The activation kinetics [of the sodium current] being fast, the variable $m$ is replaced by its equilibrium function (. . .)" [42]. However, in the original version published three years before, the activation of T-type calcium channel was not at the same timescale as the sodium channel activation [77]. In the same way for model 6, it is written "We employ a simplified version of the quantitative model originally formulated by Wang et al. (1991). Activation $s$ is rapid, and inactivation $h$ is relatively slow, (. . .) activation is assumed to be instantaneous." [43]. The similar simplification occurs for model 3 published in 1998: ten years later, the author mentioned "Note that the activation variable $s$ is considered here at steady-state, because the activation is fast compared to inactivation. This T-current model was also used with an independent activation variable ([39]), but produced very similar results as the model with activation at steady-state" [48].

Modeling the T-type calcium channel activation on the same timescale as the sodium channel activiation is a modeling simplification that does not only appears in models of thalamic neurons. For example in basal ganglia, subthalamic neurons are excitatory neurons projecting to inhibitory globus pallidus neurons. These neurons present a switch in firing activity that leads to different brain states, particularly relevant in movement generation and in Parkinson's disease [3, 20, 78–81]. Subthalamic neurons also embed T-type calcium channels. Once again, it is common to see that the activation of sodium and T-type calcium channels are modeled on the same timescale or even considered as instantaneous [49, 50, 82].

In both situations, these models are operational. It means that differentiating the sodium and T-type calcium channel activations is *not indispensable to reproduce* discharge modes. Those models are able to fire in tonic and burst. This timescale separation between the activation of these two different channels is neglected because it seems to be a computational detail. Figs 3, 5 and 7 confirm that models initially considering a fast activation of T-type calcium channel can perform the desired activity. However, our results stress the crucial importance of this timescale separation for the *robustness* in network switches. The optimum operating timescale for T-type calcium channel activation is located between the fast activation of sodium channels and the ultraslow inactivation of T-type calcium channels.

As mentioned earlier, both channels have different electrophysiological properties. Assuming the T-type calcium channel activation as fast as sodium channel activation means that it is just a current summation. This has for consequence to drastically reduce the ability of generating a robust network activity as shown in our computational experiments.

## Compatibility between switches in brain states, synaptic plasticity and neuromodulation

Neuronal oscillations, synaptic plasticity and neuromodulation are hot topics in neuroscience since they are building blocks for information processing, learning, memory or adaptability. Studying the interaction between them requires computational models that generate *robust* activity even if the synaptic connectivity and the endogenous parameters are altered.

Altogether, this paper reveals that the *robustness* in the brain state switches in presence of cellular variability, heterogeneity at the network level is *correlated with the timescale separation* between sodium and T-type calcium channel activations. Despite the quantitative differences between the models, this timescale separation makes the rhythmic transition compatible with temporal variability and spatial heterogeneity induced by regulatory functions like neuromodulation, synaptic plasticity and homeostasis. In particular, triggering oscillations without any modification of synaptic connectivity makes the models well suited to study how a change in network activity can affect learning.

The mechanism highlighted in this paper can be exploited in other models than thalamic neuron models. It is can be extended to neurons that embeds slow-activating voltage-gated calcium channels or slow-inactivating potassium channels [32]. The term slow confirms that the model should conserve the timescale distinction between the fast activation of sodium channels and the slow operating timescale of these two specified channels. This physiological timescale separation is imposed by ion channel dynamics. In conductance-based models, this corresponds to the time constant of the differential equation associated to the channel gating variable. It means that the time constant associated to sodium channel activation is an order of magnitude smaller than the time constant associated to the T-calcium channel activation.

Talking in terms of positive feedback sources can be another approach to support the importance of differentiating ion channel kinetics. On one hand, sodium channel activation acts as a *source of fast positive feedback* because depolarizing the cell drives sodium ions in,

making the cell even more depolarized and allowing more and more sodium ions to enter and so on. And the other hand, calcium channel activation operates in a similar way; it is also a source of positive feedback. However, there is a crucial dynamical difference between these two sources: they operates on two different timescales. If the two sources are acting on the same timescale, it is simply a sum of positive feedbacks. As shown by our experiments, it significantly decreases the robustness in presence of neuromodulation and synaptic plasticity. Each type of channel activation or inactivation that is a source of slow positive feedback, for example slow-activating voltage-gated calcium channels or slow-inactivating potassium channels, must be modeled an order of magnitude slower than the sodium channel activation.

Beyond the argument based on the time constant or the source of positive feedback, this time scale separation can be brought to light by dynamical analyses (see Figs 6 and 7). The slow activation of T-type calcium channels unfolds a lower branch in the V-nullcline during the bursting mode. This shape is required for a robust and tunable firing activity as shown by computational experiments [32, 34–36, 58]. Accelerating this channel activation distorts the V-nullcline and makes disappear this lower branch. The switch to bursting becomes fragile and rigid.

## Methods

All simulations were performed using the Julia programming language citation for Julia [83]. Analysis were performed either in Matlab and Excel. Code files are freely available at http://www.montefiore.ulg.ac.be/~guilldrion/Files/Jacquerie2021_codes.zip and https://osf.io/sth4d/.

### Conductance-based modeling

Single-compartment Hodgkin-Huxley models are used for all neuron models where the membrane potential, $V_m$, evolves as follows:

$$C_m \frac{dV_m}{dt} = -\sum I_i + I_{app}$$

where $C_m$ is the membrane capacitance, $I_i$ is the $i$-th current due to ionic channels, $I_{app}$ is an external applied current. Ionic currents are voltage-dependent; they are expressed as follows:

$$I_i = \bar{g}_i m_i^{p_i}(V_m) h_i^{q_i}(V_m)(V_m - E_i)$$

where $\bar{g}_i$ is the maximal conductance, $m_i$ is the activation variable, $h_i$ is the inactivation variable, $p_i$ is an integer between 1 and 4, $q_i$ is either 0 or 1, and $E_i$ is the reversal potential of the channel.

The activation and inactivation variable evolve such as:

$$\frac{dm_i}{dt} = (m_{i,\infty}(V_m) - m_i)/\tau_{m_i}(V_m)$$

$$\frac{dh_i}{dt} = (h_{i,\infty}(V_m) - h_i)/\tau_{h_i}(V_m)$$

where $m_{i,\infty}$ and $h_{i,\infty}$ are the steady-state values of the activation and inactivation variables, and $\tau_{m_i}$ and $\tau_{h_i}$ are their respective voltage-dependent time constants. Functions and parameter values are listed in S1 Supplementary Material for each model.

The computational models are run on Julia, a programming language. The ordinary differential equations (ode) are solved with an Euler explicit method. The step time is adapted depending on the model.

## Computational experiment at single-cell

Fig 1A is generated with the model 1 described in [37]. The external current, which hyperpolarizes the cell after 0.5s, switches the firing activity from tonic mode to bursting mode.

Fig 1B is constructed following the method described in [45] for model 1. The currentscape helps to visualize the contribution of each ionic current as the percentage of the total current.

Fig 1C displays traces of models 1, 5 and 5' simulated for two values of the membrane capacitance: the nominal value $C_m$ and when it is divided by 10 ($C_m/10$). The current protocol is depolarizing during 0.5s and then hyperpolarizing during 1.5s.

Fig 1D is a quantitative representation of the model capability to switch from tonic to burst for several values of the membrane capacitance $C_m$. A depolarizing current is applied during 1.5s followed by a hyperpolarizing current during 5.5s. We automatically check the rhythmic pattern; we wait for 0.5s to take into account transient effects, then we record spike times. A spike is considered when the voltage value is greater than -10mV and we save the time at which the event occurs. From spike times, we can deduce the firing pattern. A cell that has no recorded spike time is defined as *silent*. The distinction between tonic mode and bursting mode is based on the comparison of the maximal and minimal interspike-interval (ISI). A neuron is *bursting* when the maximal value of ISI is three times greater than the smallest ISI (max [ISI] > 3 min[ISI]). It ensures the cell to have clusters of action potentials separated from each other by silent intervals. If this criterion is not respected, the cell is classified as a *tonically firing* cell with regular action potential generations [37, 46]. For the nominal parameter set, the value of $I_{app}$ is known to generate tonic mode. Afterwards, the membrane capacitance $C_m$ is altered with a multiplicative factor varying from 0.01 to 0.1 with a step of 0.01 ([0.01:0.01:0.1]$C_m$), then from 0.1 to 5 with a step of 0.1 ([0.1:0.1:5] $C_m$). In addition, at each tested value of $C_m$, we scan several values $I_{app}$ in order to find the largest range of capacitance values leading to a switch from tonic to burst. The step time is also adapted for small values of the membrane capacitance to guarantee stability when solving the ode with Euler's method (for numerical values see S1 Supplementary Material).

## Computational experiment on a 2-cell circuit

Fig 2A (left) illustrates a 2-cell circuit in a typical thalamic configuration. Neuron models were connected via AMPA, GABA$_A$ and GABA$_B$ synapses. AMPA synapse provides an excitatory current while GABA is used as an inhibitory current. They are modeled as follows:

$$I_{\text{AMPA}} = \bar{g}_{\text{AMPA}}\text{AMPA}(V_m - 0)$$

$$I_{\text{GABA}_A} = \bar{g}_{\text{GABA}_A}\text{GABA}_A(V_m - V_{Cl})$$

$$I_{\text{GABA}_B} = \bar{g}_{\text{GABA}_B}\text{GABA}_B(V_m - V_K)$$

where $\bar{g}_{\text{AMPA}}$, $\bar{g}_{\text{GABA}_A}$ and $\bar{g}_{\text{GABA}_B}$ are the synaptic weights. AMPA receptor reversal potential is set to 0mV, GABA$_A$ receptor reversal potential is set to chloride reversal potential ($V_{Cl} = -70$mV) and GABA$_B$ receptor reversal potential is set to potassium reversal potential ($V_K = -85$mV). AMPA, GABA$_A$ and GABA$_B$ are variables whose dynamics depends on the

presynaptic membrane potential $V_{pre}$ following the equations

$$\dot{\text{AMPA}} = 1.1 T_m(V_{pre})[1 - \text{AMPA}] - 0.19\text{AMPA}$$

$$\dot{\text{GABA}}_\text{A} = 0.53 T_m(V_{pre})[1 - \text{GABA}_\text{A}] - 0.18\text{GABA}_\text{A}$$

$$\dot{\text{GABA}}_\text{B} = 0.016 T_m(V_{pre})[1 - \text{GABA}_\text{B}] - 0.0047\text{GABA}_\text{B}$$

with $T_m(V_{pre}) = \frac{1}{1+ \exp{[-(V_{pre}-2)/5]}}$

Synaptic weights ($\bar{g}_{syn}$ referring to $\bar{g}_{\text{AMPA}}$, $\bar{g}_{\text{GABA}_\text{A}}$ and $\bar{g}_{\text{GABA}_\text{B}}$) are initially chosen for each model in a way that the 2-cell circuit performs a rhythmic transition when the inhibitory cell is hyperpolarized. Fig 2A, center panel illustrates traces of the desired outcome (recording from model 1 with two identical cells). By contrast, Fig 2A, right panel exhibits an asynchronous state (recording from model 1 when $\bar{g}_{KCa}$ is divided by 10 and $\bar{g}_{CaT}$ is divided by 5).

Fig 2B is a quantitative comparison of model capability to switch for an increasing intrinsic variability. The intrinsic feature refers to maximal ionic conductances ($\bar{g}_i$). Each maximal conductance is randomly picked with respect to a uniform distribution in a fixed interval around its nominal value. The interval width defines the variability level, as a percentage around this nominal value. For instance, for an intrinsic variability of 10%, each ionic conductance is selected in the range: $[\bar{g}_i - 0.1\bar{g}_i, \bar{g}_i + 0.1\bar{g}_i]$. Regarding synaptic connections, each synaptic weight ($\bar{g}_{syn}$) is taken randomly with respect to uniform distribution around its nominal value in this interval $[\bar{g}_{syn} - \bar{g}_{syn}/8, \bar{g}_{syn} + \bar{g}_{syn}/8]$. For each model, one thousand sets of parameters are generated in order to build one thousand 2-cell circuits.

Each circuit is simulated during 82s. The external current depolarizes the inhibitory cell during 41s and then hyperpolarizes it during 41s. After 1s of transient period in each state, we record the spike time, *i.e.* a spike is considered when the membrane voltage is greater than -20mV. Then we identify the firing pattern of each cell based on its spike times. A cell is silent when it has not fired. A cell is bursting when the maximal interspike interval is four times greater than the minimum interspike interval, otherwise, the cell is in tonic mode. Then, we identify if the circuit is performing a rhythmic transition. During the depolarized state, the excitatory cell is silent and the inhibitory cell is spiking. During the hyperpolarized state, both cells are synchronously bursting.

## Computational experiment on a 2-cell circuit with a varying T-type calcium activation time constant

Fig 3A shows the excitatory-inhibitory 2-cell circuit affected by neuromodulation, synaptic plasticity and a tunable time constant for the T-type calcium channel activation ($\tau_{m_{CaT}}$). For each model, we generate 400 different 2-cell circuits. For each circuit, maximal intrinsic conductances and synaptic conductances are randomly picked with respect to a uniform distribution in an interval of 20% around their nominal value ($[\bar{g}_i - 20\%\bar{g}_i; \bar{g}_i + 20\%\bar{g}_i]$). These 400 circuits associated with 400 sets of conductances are simulated for varying CaT channel activation time constant; $\tau_{m_{CaT}}$ is scaled with the following multiplicative factors; [1/100, 1/50, 1/20, 1/10, 1/8, 1/5, 1/4, 1/3, 1/2, 1/1.5, 1, 1.5, 2, 3, 4, 5, 8, 10, 20, 50, 100]. For each model, we check automatically, at every scaled $\tau_{m_{CaT}}$, how many circuits among the 400 simulated have performed a rhythmic transition according the same procedure as described above.

Fig 3B summarizes the numerical values of $\tau_{m_{Na}}(V_m = V_{th})$, $\tau_{m_{CaT}}(V_m = V_{th})$ and $\tau_{h_{CaT}}(V_m = V_{th})$. Since the time constants are voltage-dependent, to compare them we fix $V_m$ equal to a threshold voltage, $V_{th}$. The threshold voltage for calcium channel activation is

chosen at the beginning of the spike upstroke. The threshold voltage for sodium channel activation is chosen at the spike initiation depending on each model. The threshold voltage for calcium channel inactivation is fixed at the beginning of the calcium spike (see S1 Supplementary Material for numerical values).

On Fig 3C, the x-axis represents the *scaled* CaT activation time constant *i.e.* the multiplicative factor of $\tau_{m_{CaT}}(V_{th})$ displayed on a logarithmic scale. In order to graduate the axis with a numerical value, the expression is evaluated at the threshold voltage $V_m = V_{th}$. The y-axis represents the percentage of rhythmic circuits. The time constants of the Na channel activation, $\tau_{m_{Na}}(V_m)$ and inactivation $\tau_{h_{CaT}}(V_m)$ are also marked on the graph with dashed vertical lines. They are evaluated at the threshold voltage $V_m = V_{th}$. The results are robust to the choice of the threshold voltage. Models 5' and 6' have an instantaneous sodium activation channel, therefore the right boundary is replaced by $\tau_{m_{CaT}}/100$.

Fig 3D combines the individual model robustness analysis on one graph. Since channel activation or inactivation time constants have not the same order of magnitude between each model, the x-axis is a *normalized* logarithmic scale. It is normalized such as the left (resp. right) boundary is the time constant of the Na channel activation (resp. CaT channel inactivation) evaluated at its threshold voltage. We scale the axis with a logarithmic scale such as $\frac{\log(\tau_{m_{CaT}}(V_{th})) - \log(\tau_{m_{Na}}(V_{th}))}{\log(\tau_{h_{CaT}}(V_{th})) - \log(\tau_{m_{Na}}(V_{th}))}$ where $\tau_{m_{CaT}}(V_{th})$ is affected by the multiplicative factor. The idea was to superpose every model between their respective fast timescale (associated with $\tau_{m_{Na}}$) and ultraslow timescale (associated with $\tau_{h_{CaT}}$). It corresponds to scale each individual plot from Fig 3C in the window bounded by $\tau_{m_{Na}}(V_{th})$ and $\tau_{h_{CaT}}(V_{th})$.

## Computational experiment on a 200-cell network

Fig 4A (resp. C) displays the network configuration and the connectivity between the excitatory and inhibitory neuron populations for a homogeneous (resp. heterogeneous) network. E-cells are connected to I-cells via AMPA synapses and I-cells are connected to E-cells via GABA$_A$ and GABA$_B$. A homogeneous network is built with 200 identical neurons, and the synaptic weights are the same between each neuron. By contrast, a heterogeneous network is built with neuron models whose maximal ionic conductances are randomly picked in an interval whose width is model-dependent. The interval width is equal to 20% around their nominal values for models 1,2,3,5', 10% for models 4 and 6' and 5% for model 5. Synaptic weights are taken randomly with respect to uniform distribution around its nominal value in this interval $[\bar{g}_{syn} - \bar{g}_{syn}/8, \bar{g}_{syn} + \bar{g}_{syn}/8]$.

Fig 4B and 4D display a zoom of the the voltage-traces from two I-cells of the population during 4s, starting 500ms before the switch(recording from model 1—left column, and model 5 -right column). Analyzing activity in a large neuron population is performed through the computation of the local field potential (LFP). It is the mean field measure of the average behavior of interacting neurons [1, 84, 85]. Fig 4B and 4D (third curves) reflect the collective synaptic activity of the neuronal population. It is modeled by the normalized sum of the post-synaptic currents: LFP $= \frac{1}{M}\sum_{j=1}^{M} I_{syn,j}$ where $M$ is the number of post-synaptic neurons in the population. The post-synaptic current from neuron $i$ to neuron $j$ is

$$I_{syn,ij} = -g_{k,i}(V_m - V_k)$$

where $k$ is the receptor type (AMPA, GABA$_A$ and GABA$_B$). The entire post synaptic current of the neuron $j$ is the sum of the post-synaptic current for all the neighboring pre-synaptic

neurons. The sum over the $N$ neuron is computed:

$$I_{syn,j} = \frac{1}{N} \sum_{i=1}^{N} I_{syn,ij}$$

where $N$ is the number of pre-synaptic neurons to the neuron $j$. Finally, to compute the local field potential, the sum of all the post-synaptic current of all the neurons are given by:

$$LFP = \frac{1}{M} \sum_{j=1}^{M} I_{syn,j}$$

where $M$ is the number of post-synaptic neurons in the population.

The time-course of LFP traces shows a oscillating pattern when neurons switch in synchronous bursting. This comes from the large synaptic currents during a burst times the number of neurons in the large population of 200 cells compared to the single input during a spike. To analyse the frequency content of this oscillating trace, LFPs are low-pass filtered at 100Hz via a fourth order Butterworth filter reflecting the use of macro-electrodes in LFP acquisition. The time-frequency plot shows the evolution of the frequency content along time. It results from a logarithmic representation of the spectrogram LFP obtained via a short-time Fourier transform [37]. Spectrograms are obtained via Matlab function `spectrogram` and the input parameters such as the sampling frequency, the time window, the overlap period, the signal-noise-ration (SNR) are adapted for each model (see S1 Supplementary Material. for numerical values).

Fig 4B and 4D illustrate the spectrogram of the LFP of the inhibitory neuron population for model 1 (resp. model 5) on the left (resp. right) panel. The homogeneous and the heterogeneous network are respectively on top and bottom. The simulation is performed during 42s split into 21s of depolarized state followed by 21s of hyperpolarized state. When the network is in oscillatory state, the spectrogram is marked by a high power LFP frequency band (yellow band) characterizing that the mean-field rhythmic activity is turned on.

## Computational experiment on a 200-cell network with a varying T-type calcium activation time constant

Fig 5 reproduces the 200-cell network of the Fig 4 with the same current protocol. The 200-cell network is built by randomly picked intrinsic and synaptic conductances in a given interval around their nominal values. The width of the interval is successively increased with steps of 5% around the nominal values (from 0% to 50%). At each variability order, the heterogeneous network sees the CaT activation time constant $\tau_{m_{CaT}}(V_m)$ of every neuron scaled by a multiplicative factor equal to [1/8, 1/5, 1/4, 1/3, 1/2, 1, 2, 3, 4, 5, 8]. The LFP activity of the neuron population is plotted similarly as in Fig 4. At each scaled time constant, we detect if the mean-field activity is turned on by analyzing the time course and the spectrogram of the LFP. If the time-course shows a transition in active state to an oscillatory state and if the spectrogram is marked by a high power band during the hyperpolarized state, the network is considered to switch. We summarize the computational experiment in a table showing vertically the different multiplicative factor and the color shows the largest width of the variability interval at which the network switches.

## Construction of the reduced models and phase portrait analysis

We reduce high-dimensional conductance-based models into a minimal model built with three variables: the membrane voltage $V$, the slow variable $V_s$ and the ultraslow variable $V_u$.

We chose on purpose to write $V_m$ for the membrane voltage in conductance based models and $V$ for the membrane voltage variable in reduced models to insist on the modeling difference. The dimensionality reduction is performed in a rigorous and systematic manner.

The reduced model is described by three differential equations:

$$
\begin{aligned}
C_m dV/dt &= -\sum I_i + \mathrm{I_{app}} \\
dV_s/dt &= (V - V_s)/\tau_s(V) \\
dV_u/dt &= (V - V_u)/\tau_u(V)
\end{aligned}
$$

The slow (resp. ultraslow) variable is a filtered version of the membrane voltage whose time constant is voltage dependent defined as $\tau_s(V)$ (resp. $\tau_u(V)$). The activation of potassium channel (resp. inactivation of T-type calcium channel) paces the slow (resp. ultraslow) timescale $ie.$ $\tau_s(V) = \tau_{m_K}(V)$ and $\tau_u(V) = \tau_{h_{CaT}}(V)$. The fast timescale is governed by the activation of sodium channel ($\tau_f(V) = \tau_{m_{Na}}(V)$). Model 5,5',6 and 6' have considered the activation this channel as instantaneous, therefore there is no differential equation and no time constant for this activation. In those models, the fast timescale is governed by the activation of potassium channel divided by 10: $\tau_f(V) = \tau_{m_K}(V)/10$. Increasing the division factor does not change our results.

Each ionic current is expressed in terms of activation and inactivation variables (resp. $m_i$ and $h_i$). By contrast with the conductance-based models, these variables are no more computed through differential equations (see section 1 in Methods). The idea is to decompose the contribution of each ionic current into the three timescales [33]. In model reduction, it is common to assume that an activation or an inactivation variable is globally evolving on a given timescale. Here, we perform an evaluation of the activation and inactivation kinetics at each membrane voltage.

$$
I_i = \bar{g}_i m_i^{p_i}(V, V_s, V_u) h_i^{q_i}(V, V_s, V_u)(V - E_i)
$$

These activation and inactivation variables are decomposed into a weighted sum of three terms to account for their contribution in each timescale ($m_i$ and $h_i$ are contracted into $X$ to ease the reading):

$$
\begin{aligned}
X(V, V_s, V_u) = \quad & w_{fs}^X(V) & X_\infty(V) & \quad \text{fast} \\
+ \quad & (w_{su}^X(V) - w_{fs}^X(V)) & X_\infty(V_s) & \quad \text{slow} \\
+ \quad & (1 - w_{su}^X(V)) & X_\infty(V_u) & \quad \text{ultraslow}
\end{aligned}
$$

The two voltage-dependent weighting factors $w_{fs}^X(V)$ $w_{su}^X(V)$ of the activation or inactivation variable are obtained by comparing its associated time constant defined in the conductance-based model ($\tau_X(V)$) at a given membrane voltage with respect to the three time

constants pacing each timescale ($\tau_f(V)$, $\tau_s(V)$ and $\tau_u(V)$).

$$\text{if } \tau_X(V) \leq \tau_f(V)$$

$$w_{fs}^X(V) = 1$$

$$w_{su}^X(V) = 1$$

$$\text{else if } \tau_f(V) < \tau_X(V) \leq \tau_s(V)$$

$$w_{fs}^X(V) = \frac{\log(\tau_s(V)) - \log(\tau_X(V))}{\log(\tau_s(V)) - \log(\tau_f(V))}$$

$$w_{su}^X(V) = 1$$

$$\text{else if } \tau_s(V) < \tau_X(V) \leq \tau_u(V)$$

$$w_{fs}^X(V) = 0$$

$$w_{su}^X(V) = \frac{\log(\tau_u(V)) - \log(\tau_X(V))}{\log(\tau_u(V)) - \log(\tau_s(V))}$$

$$\text{else if } \tau_X(V) > \tau_u(V)$$

$$w_{fs}^X(V) = 0$$

$$w_{su}^X(V) = 0$$

The weights are either 0,1 or equal to the logarithmic distance between the referential time constants (for more explanations about the algorithm see [33]).

Fig 6 shows the voltage trace of the reduced model 1 and its associated phase portrait at two given instants indicated by 1 and 2. Each column corresponds to a reduction under three conditions imposed on the time constant of the T-type calcium channel activation $\tau_{m_{CaT}}(V)$; $\tau_{m_{CaT}}(V)/50$ (left), $\tau_{m_{CaT}}(V)$ (center) and $50\tau_{m_{CaT}}(V)$ (right). This modification has an effect on the weighted sum association to the $m_{CaT}(V, V_s, V_u)$ since the contribution of this channel into the three timescales is affected by the multiplicative factor.

The phase plane analysis is performed on Matlab. The V-nullcline is defined by $dV/dt = 0$ giving $0 = (1/C_m)(-\sum m_i^{p_i}(V, V_s, V_u)h_i^{q_i}(V, V_s, V_u)(V - E_i) + I_{app})$. The Vs-nullcline is equal to $0 = (V - V_s)/\tau_s(V)$ leading to $V_s = V$ (as shown by the straight line in the different phase portraits). In the tonic mode, the V-nullcline is computed at a given time during the depolarized state. In the bursting mode, the V-nullcline is changing its shape throughout the burst generation. It mimics the physiological activation and inactivation of the T-type calcium current and in the reduced model it corresponds to oscillation of the ultraslow variable (see [58] for more explanations). Therefore, to compare the different phase portrait between the different models and the different CaT time constants, we chose to draw the phase portrait at the saddle node bifurcation. By definition, at the saddle node bifurcation, the two nullclines are intersecting each other and the determinant of the jacobian must be equal to zero. To

obtain the value of $V_u$ at the saddle node bifurcation, the system to solve is equal to:

$$0 \quad = -\sum I_i + \mathrm{I_{app}}$$

$$0 \quad = (V - V_s)/\tau_s(V)$$

$$0 \quad = \begin{vmatrix} \dfrac{\partial(dV/dt)}{\partial V} & \dfrac{\partial(dV/dt)}{\partial V_s} \\[2ex] \dfrac{\partial(dV_s/dt)}{\partial V} & \dfrac{\partial(dV_s/dt)}{\partial V_s} \end{vmatrix}$$

When there is no saddle-node bifurcation, the phase portrait is drawn at the fixed point computed by: $dV/dt = 0$, $dV_s/dt = 0$, $dV_u/dt = 0$. The membrane voltage is indeed attracted by a hyperpolarized stable fixed point and remains silent.

Fig 7 is obtained in a similar manner as Fig 6 except that the membrane capacitance $C_m$ is scaled by a factor of 1/3 (right column). The phase portraits are drawn at the saddle-node bifurcation. The algorithm to compute the V-nullcline is the same. The trajectory is added on the phase plane to illustrate the first action potential from the hyerpolarized state towards the limit cycle. As shown in S4 Video the V-nullcline is changing its shape. The scaling factor only affects the first equation $dV/dt = (3/C_m)(-\Sigma I_i + I_{app})$. The velocity in the horizontal axis is increased by 3. It is clearly shown by the trajectory profile that is stronger attracted along the x-axis.

More information concerning the model reduction is available on Julia and Matlab codes. The numerical values for each reduced model are given in S1 Supplementary Material.

## Supporting information

**S1 Video. Membrane voltage evolution during a hyperpolarized-induced bursting (top) and its associated phase portrait considering a *fast* activation of T-type calcium channel (bottom).** V-(resp. Vs-) nullcline is skteched in blue (resp. green). The trajectory is marked by round circles. The simulation is shown for the reduced model 1 (Video associated to Fig 6 (left)).
(MP4)

**S2 Video. Membrane voltage evolution during a hyperpolarized-induced bursting (top) and its associated phase portrait for a *slow* activation of T-type calcium channel (bottom).** V-(resp. Vs-) nullcline is skteched in blue (resp. green). The trajectory is marked by round circles. The simulation is shown for the reduced model 1 (Video associated to Fig 6 (center)).
(MP4)

**S3 Video. Membrane voltage evolution during a hyperpolarized-induced bursting (top) and its associated phase portrait for a *ultraslow* activation of T-type calcium channel (bottom).** V-(resp. Vs-) nullcline is skteched in blue (resp. green). The trajectory is marked by round circles. The simulation is shown for the reduced model 1 (Video associated to Fig 6 (right)).
(MP4)

**S4 Video. Phase portrait evolution as a function of the T-type calcium channel activation kinetics.** V-(resp. Vs-) nullcline is skteched in blue (resp. green). The multiplicative factor of the T-type calcium channel activation time constant is indicated by $\eta$. Apparition of the lower branch in the V-nullcline when the activation decelerates. Then, the lower branch disappears

into a hourglass shape. Results shown for the reduced model 1. Similar phase portraits evolution of the reduced models 2, 5' and 6' are available on http://www.montefiore.ulg.ac.be/~guilldrion/Files/Jacquerie2021_codes.zip (in the folder "video").
(MP4)

**S5 Video. Membrane voltage evolution during a hyperpolarized-induced bursting (top) and its associated phase portrait at the nominal value of the membrane capacitance (bottom).** V-(resp. Vs-) nullcline is skteched in blue (resp. green). The trajectory is marked by round circles (Video associated to Fig 7A (left)).
(MP4)

**S6 Video. Membrane voltage evolution during a hyperpolarized-induced bursting (top) and its associated phase portrait when the membrane capacitance is scaled by 1/3 (bottom).** V-(resp. Vs-) nullcline is skteched in blue (resp. green). The trajectory is marked by round circles (Video associated to Fig 7A (right)).
(MP4)

**S7 Video. Membrane voltage evolution during a hyperpolarized-induced bursting (top) and its associated phase portrait at the nominal value of the membrane capacitance (bottom).** V-(resp. Vs-) nullcline is skteched in blue (resp. green). The trajectory is marked by round circles (Video associated to Fig 7B (left)).
(MP4)

**S8 Video. Membrane voltage evolution during a hyperpolarized-induced bursting (top) and its associated phase portrait when the membrane capacitance is scaled by 1/3 (bottom).** V-(resp. Vs-) nullcline is skteched in blue (resp. green). The trajectory is marked by round circles (Video associated to Fig 7B (right)).
(MP4)

**S1 Supplementary Material. A.** : Quantification of the firing pattern properties. **B**: Simulation of the reduced models not exhibited in the Results. **C**: Model description and their parameter values. **D**: Ionic channel description: steady-state functions and time constants of the gating variables. **E**: Description of reduced models and their parameter values.
(PDF)

## Acknowledgments

The authors gratefully acknowledge Leandro M. Alonso, from Volen Center and Department of Biology in Brandeis University, for his python codes used for the currentscape generation.

## Author Contributions

**Conceptualization:** Guillaume Drion.

**Formal analysis:** Kathleen Jacquerie.

**Methodology:** Kathleen Jacquerie, Guillaume Drion.

**Resources:** Guillaume Drion.

**Software:** Kathleen Jacquerie.

**Supervision:** Guillaume Drion.

**Visualization:** Kathleen Jacquerie, Guillaume Drion.

**Writing – original draft:** Kathleen Jacquerie.

**Writing – review & editing:** Kathleen Jacquerie, Guillaume Drion.

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
