## [Decision Letter · Decision Letter 0]

23 Jan 2021

Dear Jacquerie,

Thank you very much for submitting your manuscript "Robust switches in thalamic network activity require a timescale separation between sodium and T-type calcium channel activations" for consideration at PLOS Computational Biology.

As with all papers reviewed by the journal, your manuscript was reviewed by members of the editorial board and by several independent reviewers. In light of the reviews (below this email), we would like to invite the resubmission of a significantly-revised version that takes into account the reviewers' comments.

We cannot make any decision about publication until we have seen the revised manuscript and your response to the reviewers' comments. Your revised manuscript is also likely to be sent to reviewers for further evaluation.

Sincerely,

Hugues Berry

Associate Editor

PLOS Computational Biology

Samuel Gershman

Deputy Editor

PLOS Computational Biology

Reviewer's Responses to Questions

**Comments to the Authors:**

Reviewer #1: Comments on the manuscript PCOMPBIOL-D-20-01928 by Jacquerie K and Drion G.

The authors study a possible contribution of the activation kinetics of the low threshold calcium current IT to the robustness of subthreshold oscillations of models of thalamic neurons. The study is justified based on the observation that two previously published models of thalamocortical neurons, in which the activation variable of IT is considered instantaneous, are less robust to parameter variations than models that consider relatively slow, more physiologically plausible, IT activation kinetics to switch between tonic and bursting modes. The effect of changing the time scale of IT activation on the robustness of firing mode switching is systematically explored at the cellular and network levels. Robustness of a given model is, in turn, evaluated for its ability to undergo oscillations after variation of parameters such as global conductance scaling (capacitance variation), several degrees of intrinsic and synaptic conductance variability and homogeneity or heterogeneity of connectivity of an artificial network.

The results show that cell and network oscillatory robustness requires a T type calcium current that activates with kinetics between those of its very slow inactivation and the fast activation of the sodium current. Base on this finding, the authors speculate with a more general principle in which the robustness of transitions between different network rhythms (and consequently different functional states) requires both fast and slow positive feedback mechanisms with time scale differences of about one order of magnitude.

The study is well organized and is clearly presented. I personally appreciated the way the sequential increase in complexity of the computational experiments and results is presented; this facilitates comprehension and readability of the manuscript. In general, I consider this to be a good piece of theoretical work that supports the notion that the activation of IT should be modeled preserving the time dependence that has been determined experimentally. Results presented in figure 3C in which the maximum percentage of rhythmic networks coincide almost perfectly with the “natural” (multiplicative factor of 1) kinetics of τmCAT are striking. As pointed out by the authors, reduction of complex relatively fast processes to instantaneous, while maintaining certain phenomenological accuracy, is computationally convenient and also facilitates the mathematical analysis of neuronal models. However, the results presented by Jacquerie and Drion indicate that important properties such as the robustness of oscillatory mode switching could be compromised by this simplification. It is also possible that the dynamical properties of cells and networks could differ if these reductions are made. In this regard, although is out of the scope of the study, it would be interesting to perform an analysis, similar to that performed in the cited work by Rush and Rinzel (model 6 in the present work), to see if the dynamical (bifurcation) structure still holds after including a non-instantaneous activation of IT (e.g. model 6’ in the present work).

I do not have any substantial comment on the quality, originality and importance of the study. I only suggest that a comment should be included in the discussion about the physiological relevance of the findings. It is known that T type channels are subject to modulation by several signaling cascades and are also targets of several experimental and clinical drugs currently in use. It opens the possibility that modulations that change activation kinetics of T type channels have important consequences on physiological and pathological states (e.g. sleep and epilepsy respectively) that involve oscillations generated, sustained or propagated by the thalamic circuit.

Reviewer #2: In this study, the authors examined 6 different computational models of thalamocortical neurons, in order to understand common property which is required for switching between post-inhibitory bursting and tonic states. The time scale separation in sodium and calcium currents was identified as a important property across models. They also identified that robustness of this switching, measured by varying parameters, is higher when kinetics were included in the computational model. While the finding on the common property across models involving time scale separation add incremental value to understanding computational models, the finding about robustness does not seem to have clear implications. Below are main issues which could be addressed, to justify the need for publication in this journal.

1. One of the main result of this study is the demonstration of time scale separation in the calcium and Na channel as an important mechanism across models for switching between bursting and tonic states. Previous models have used this mechanism for the switching, and this study help extend the importance of this property across models. In the same motivation, it would be helpful to obtain a minimal model that include all the required currents for switching. Further, it would be helpful to describe and classify common biologically relevant processes, eg. progressive buildup of L current is common across all models.

2. It would be helpful to examine the time scale separation from dynamical system point. Specifically, authors could add simple phase space plots of the two currents that shows similar qualitative behavior across different models.

3. With respect to results on robustness, authors first show an increase in robustness when kinetics of is implemented compared to instantaneous assumptions for Ca currents. This finding is not surprising. Instantaneous activation is often chosen to have better understanding of other variables or to improve speed under fixed parameter regime. It is not clear what the authors like to achieve in demonstrating improved robustness with adding dynamical variable to instantaneous assumptions, since the outcome is expected.

4. There are some differences in robustness across models (that include kinetics) but the study does not provide no additional understanding of reason for this differences. TO name a few, does it arise from higher coupling between slow currents ?, do H-current play additional role in improved robustness ? is the phase space larger for the each state or a difference in way the switching happens?

5. Do other property of the bursting and firing rate (like number of burst vary within the regions of switching) ? How much is the variation in these properties ?

6. Robustness is helpful in maintain similar qualitative behavior under small changes in neuromodulation or synaptic plasticity. But, it also critical to have neuromodulation change the switching states, such as between awake/sleep transition. It would be useful to identify boundaries of small and large variations.

7. To study robustness, why was Cm chosen as the parameter ?, why not other parameters ?

8. I or RE cells have many intrinsic properties as well, it is not clear if these were examined.

Reviewer #3: review is uploaded as an attachment

**Have all data underlying the figures and results presented in the manuscript been provided?**

Reviewer #1: Yes

Reviewer #2: Yes

Reviewer #3: Yes

PLOS authors have the option to publish the peer review history of their article (what does this mean?). If published, this will include your full peer review and any attached files.

Reviewer #1: No

Reviewer #2: No

Reviewer #3: **Yes: **Leandro M. Alonso
---

## [Decision Letter · Decision Letter 1]

23 Apr 2021

Dear Jacquerie,

We are pleased to inform you that your manuscript 'Robust switches in thalamic network activity require a timescale separation between sodium and T-type calcium channel activations' has been provisionally accepted for publication in PLOS Computational Biology.

Best regards,

Hugues Berry

Associate Editor

PLOS Computational Biology

Samuel Gershman

Deputy Editor

PLOS Computational Biology

Reviewer's Responses to Questions

**Comments to the Authors:**

Reviewer #1: All my scientific concerns have been adequately addressed. However, there are a few minor mistakes that should be corrected. Line numbers refer to those in the revised version with changes highlighted.

1. Figure 1 (figure legend) remove “green models” since that convention was removed from the main text.

2. Figure 2 (figure legend) I guess “proportion in percentage of” is redundant. Change it for “percentage of” instead

3. Line 381: “helping him to operate”. Should be: “helping it to operate”

4. Line 404: “to obtain a three dimensional models”. Should be: “to obtain three dimensional models”

Reviewer #2: Authors addressed all my main concerns.

Reviewer #3: I thank the authors for addressing all my comments. I believe the manuscript was improved, and that the fixed figures make some of the results clearer.

**Have the authors made all data and (if applicable) computational code underlying the findings in their manuscript fully available?**

Reviewer #1: Yes

Reviewer #2: Yes

Reviewer #3: Yes

PLOS authors have the option to publish the peer review history of their article (what does this mean?). If published, this will include your full peer review and any attached files.

Reviewer #1: No

Reviewer #2: **Yes: **Giri P Krishnan

Reviewer #3: **Yes: **Leandro Alonso

---

## [Editor Report · Acceptance letter]

13 May 2021

PCOMPBIOL-D-20-01928R1 

Robust switches in thalamic network activity require a timescale separation between sodium and T-type calcium channel activations

Dear Dr Jacquerie,

I am pleased to inform you that your manuscript has been formally accepted for publication in PLOS Computational Biology. Your manuscript is now with our production department and you will be notified of the publication date in due course.

With kind regards,

Zsofi Zombor
